# Explaining Landscape Connectivity of Low-cost Solutions for Multilayer Nets

**Rohith Kuditipudi**
Duke University
rohith.kuditipudi@duke.edu

**Xiang Wang**
Duke University
xwang@cs.duke.edu

**Holden Lee**
Princeton University
holdenl@princeton.edu

**Yi Zhang**
Princeton University
y.zhang@cs.princeton.edu

**Zhiyuan Li**
Princeton University
zhiyuanli@cs.princeton.edu

**Wei Hu**
Princeton University
huwei@cs.princeton.edu

**Sanjeev Arora**
Princeton University and Institute for Advanced Study
arora@cs.princeton.edu

**Rong Ge**
Duke University
rongge@cs.duke.edu

## Abstract

*Mode connectivity* (Garipov et al., 2018; Draxler et al., 2018) is a surprising phenomenon in the loss landscape of deep nets. Optima—at least those discovered by gradient-based optimization—turn out to be connected by simple paths on which the loss function is almost constant. Often, these paths can be chosen to be piece-wise linear, with as few as two segments.

We give mathematical explanations for this phenomenon, assuming generic properties (such as dropout stability and noise stability) of well-trained deep nets, which have previously been identified as part of understanding the generalization properties of deep nets. Our explanation holds for realistic multilayer nets, and experiments are presented to verify the theory.

## 1 Introduction

Efforts to understand how and why deep learning works have led to a focus on the *optimization landscape* of the training loss. Since optimization to near-zero training loss occurs for many choices of random initialization, it is clear that the landscape contains many global optima (or near-optima). However, the loss can become quite high when interpolating between found optima, suggesting that these optima occur at the bottom of "valleys" surrounded on all sides by high walls. Therefore the phenomenon of *mode connectivity* (Garipov et al., 2018; Draxler et al., 2018) came as a surprise: optima (at least the ones discovered by gradient-based optimization) are connected by simple paths in the parameter space, on which the loss function is almost constant. In other words, the optima are not walled off in separate valleys as hitherto believed. More surprisingly, the paths connecting discovered optima can be piece-wise linear with as few as two segments.

Mode connectivity begs for theoretical explanation. One paper (Freeman and Bruna, 2016) attempted such an explanation for 2-layer nets, even before the discovery of the phenomenon in multilayer nets. However, they require the width of the net to be exponential in some relevant parameters. Others (Venturi et al., 2018; Liang et al., 2018; Nguyen et al., 2018; Nguyen, 2019) require special structure in their networks where the number of neurons needs to be greater than the number of training data points. Thus it remains an open problem to explain mode connectivity even in the 2-layer case with realistic parameter settings, let alone for standard multilayer architectures.

At first sight, finding a mathematical explanation of the mode connectivity phenomenon for multilayer nets—e.g., for a 50-layer ResNet on ImageNet—appears very challenging. However, the glimmer of hope is that since the phenomenon exists for a variety of architectures and datasets, it must arise from some generic property of trained nets. The fact that the connecting paths between optima can have as few as two linear segments further bolsters this hope.

Strictly speaking, empirical findings such as in (Garipov et al., 2018; Draxler et al., 2018) do not show connectivity between *all* optima, but only for *typical* optima discovered by gradient-based optimization. It seems an open question whether connectivity holds for all optima in overparametrized nets. Section 5 answers this question, via a simple example of an overparametrized two-layer net, not all of whose optima are connected via low-cost paths.

Thus to explain mode connectivity one must seek generic properties that hold for optima obtained via gradient-based optimization on realistic data. A body of work that could be a potential source of such generic properties is the ongoing effort to understand the generalization puzzle of over-parametrized nets—specifically, to understand the "true model capacity". For example, Morcos et al. (2018) note that networks that generalize are insensitive to linear restrictions in the parameter space. Arora et al. (2018) define a *noise stability* property of deep nets, whereby adding Gaussian noise to the output of a layer is found to have minimal effect on the vector computed at subsequent layers. Such properties seem to arise in a variety of architectures purely from gradient-based optimization, without any explicit noise-injection during training—though of course using small-batch gradient estimates is an implicit source of noise-injection. (Sometimes training also explicitly injects noise, e.g. dropout or batch-normalization, but that is not needed for noise stability to emerge.)

Since resilience to perturbations arises in a variety of architectures, such resilience counts as a "generic" property for which it is natural to prove mode connectivity as a consequence. We carry this out in the current paper. Note that our goal here is not to explain every known detail of mode connectivity, but rather to give a plausible first-cut explanation.

First, in Section 3 we explain mode connectivity by assuming the network is trained via dropout. In fact, the desired property is weaker: so long as there *exists* even a *single* dropout pattern that keeps the training loss close to optimal on the two solutions, our proof constructs a piece-wise linear path between them. The number of linear segments grows linearly with the depth of the net.

Then, in Section 4 we make a stronger assumption of noise stability along the lines of Arora et al. (2018) and show that it implies mode connectivity using paths with 10 linear segments. While this assumption is strong, it appears to be close to what is satisfied in practice. (Of course, one could explicitly train deep nets to satisfy the needed noise stability assumption, and the theory applies directly to them.)

## 1.1 Related work

The landscape of the loss function for training neural networks has received a lot of attention. Dauphin et al. (2014); Choromanska et al. (2015) conjectured that local minima of multi-layer neural networks have similar loss function values, and proved the result in idealized settings. For linear networks, it is known (Kawaguchi, 2016) that all local minima are also globally optimal.

Several theoretical works have explored whether a neural network has spurious valleys (non-global minima that are surrounded by other points with higher loss). Freeman and Bruna (2016) showed that for a two-layer net, if it is sufficiently overparametrized then all the local minimizers are (approximately) connected. However, in order to guarantee a small loss along the path they need the number of neurons to be exponential in the number of input dimensions. Venturi et al. (2018) proved that if the number of neurons is larger than either the number of training samples or the intrinsic dimension (infinite for standard architectures), then the neural network cannot have spurious valleys. Liang et al. (2018) proved similar results for the binary classification setting. Nguyen et al. (2018); Nguyen (2019) relaxed the requirement on overparametrization, but still require the output layer to have more direct connections than the number of training samples.

Some other papers have studied the existence of spurious local minima. Yun et al. (2018) showed that in most cases neural networks have spurious local minima. Note that a local minimum need only have loss no larger than the points in its neighborhood, so a local minimum is not necessarily a spurious valley. Safran and Shamir (2018) found spurious local minima for simple two-layer neural networks

under a Gaussian input distribution. These spurious local minima are indeed spurious valleys as they have positive definite Hessian.

## 2 Preliminaries

**Notations**    For a vector $v$, we use $\|v\|$ to denote its $\ell_2$ norm. For a matrix $A$, we use $\|A\|$ to denote its operator norm, and $\|A\|_F$ to denote its Frobenius norm. We use $[n]$ to denote the set $\{1, 2, \ldots, n\}$. We use $I_n$ to denote the identity matrix in $\mathbb{R}^{n \times n}$. We use $O(\cdot), \Omega(\cdot)$ to hide constants and use $\widetilde{O}(\cdot), \widetilde{\Omega}(\cdot)$ to hide poly-logarithmic factors.

**Neural network**    In most of the paper, we consider fully connected neural networks with ReLU activations. Note however that our results can also be extended to convolutional neural networks (in particular, see Remark 1 and the experiments in Section 6).

Suppose the network has $d$ layers. Let the vector before activation at layer $i$ be $x^i$, $i \in [d]$, where $x^d$ is just the output. For convenience, we also denote the input $x$ as $x^0$. Let $A_i$ be the weight matrix at $i$-th layer, so that we have $x^i = A_i \phi(x^{i-1})$ for $2 \leq i \leq d$ and $x^1 = A_1 x^0$. For any layer $i$, $1 \leq i \leq d$, let the width of the layer be $h_i$. We use $[A_i]_j$ to denote the $j$-th column of $A_i$. Let the maximum width of the hidden layers be $h_{\max} := \max\{h_1, h_2, \ldots, h_{d-1}\}$ and the minimum width of the hidden layers be $h_{\min} := \min\{h_1, h_2, \ldots, h_{d-1}\}$.

We use $\Theta$ to denote the set of parameters of neural network, and in our specific model, $\Theta = \mathbb{R}^{h_1 \times h_0} \times \mathbb{R}^{h_2 \times h_1} \times \cdots \times \mathbb{R}^{h_d \times h_{d-1}}$ which consists of all the weight matrices $\{A_i\}$'s.

Throughout the paper, we use $f_\theta, \theta \in \Theta$ to denote the function that is computed by the neural network. For a data set $(x, y) \sim \mathcal{D}$, the loss is defined as $L_{\mathcal{D}}(f_\theta) := \mathbb{E}_{(x,y) \sim \mathcal{D}}[l(y, f_\theta(x))]$ where $l$ is a loss function. The loss function $l(y, \hat{y})$ is convex in the second parameter. We omit the distribution $\mathcal{D}$ when it is clear from the context.

**Mode connectivity and spurious valleys**    Fixing a neural network architecture, a data set $\mathcal{D}$ and a loss function, we say two sets of parameters/solutions $\theta^A$ and $\theta^B$ are $\epsilon$-**connected** if there is a path $\pi(t) : \mathbb{R} \to \Theta$ that is continuous with respect to $t$ and satisfies: 1. $\pi(0) = \theta^A$; 2. $\pi(1) = \theta^B$ and 3. for any $t \in [0, 1]$, $L(f_{\pi(t)}) \leq \max\{L(f_{\theta^A}), L(f_{\theta^B})\} + \epsilon$. If $\epsilon = 0$, we omit $\epsilon$ and just say they are connected.

If all local minimizers are connected, then we say that the loss function has the **mode connectivity property**. However, as we later show in Section 5, this property is very strong and is not true even for overparametrized two-layer nets. Therefore we restrict our attention to classes of low-cost solutions that can be found by the gradient-based algorithms (in particular in Section 3 we focus on solutions that are dropout stable, and in Section 4 we focus on solutions that are noise stable). We say the loss function has $\epsilon$-**mode connectivity property** with respect to a class of low-cost solutions $\mathcal{C}$, if any two minimizers in $\mathcal{C}$ are $\epsilon$-connected.

Mode connectivity is closely related to the notion of spurious valleys and connected sublevel sets (Venturi et al., 2018). If a loss function has all its sublevel sets ($\{\theta : L(f_\theta) \leq \lambda\}$) connected, then it has the mode connectivity property. When the network only has the mode connectivity property with respect to a class of solutions $\mathcal{C}$, as long as the class $\mathcal{C}$ contains a global minimizer, we know there are no spurious valleys in $\mathcal{C}$.

However, we emphasize that neither mode connectivity or lack of spurious valleys implies any local search algorithm can efficiently find the global minimizer. These notions only suggest that it is unlikely for local search algorithms to get completely stuck.

## 3 Connectivity of dropout-stable optima

In this section we show that *dropout stable* solutions are connected. More concretely, we define a solution $\theta$ to be $\epsilon$-dropout stable if we can remove a subset of half its neurons in each layer such that the loss remains steady.

**Definition 1.** *(Dropout Stability) A solution $\theta$ is $\epsilon$-**dropout stable** if for all $i$ such that $1 \leq i < d$, there exists a subset of at most $\lfloor h_j/2 \rfloor$ hidden units in each of the layers $j$ from $i$ through $d - 1$ such*

*that after rescaling the outputs of these hidden units (or equivalently, the corresponding rows and/or columns of the relevant weight matrices) by some factor $r$[1] and setting the outputs of the remaining units to zero, we obtain a parameter $\theta_i$ such that $L(f_{\theta_i}) \le L(f_\theta) + \epsilon$.*

Intuitively, if a solution is $\epsilon$-dropout stable then it is essentially only using half of the network's capacity. We show that such solutions are connected:

**Theorem 1.** *Let $\theta^A$ and $\theta^B$ be two $\epsilon$-dropout stable solutions. Then there exists a path in parameter space $\pi : [0,1] \to \Theta$ between $\theta^A$ and $\theta^B$ such that $L(f_{\pi(t)}) \le \max\{L(f_{\theta^A}), L(f_{\theta^B})\} + \epsilon$ for $0 \le t \le 1$. In other words, letting $\mathcal{C}$ be the set of solutions that are $\epsilon$-dropout stable, a ReLU network has the $\epsilon$-mode connectivity property with respect to $\mathcal{C}$.*

Our path construction in Theorem 1 consists of two key steps. First we show that we can rescale at least half the hidden units in both $\theta^A$ and $\theta^B$ to zero via continuous paths of low loss, thus obtaining two parameters $\theta_1^A$ and $\theta_1^B$ satisfying the criteria in Definition 1.

**Lemma 1.** *Let $\theta$ be an $\epsilon$-dropout stable solution and let $\theta_i$ be specified as in Definition 1 for $1 \le i < d$. Then there exists a path in parameter space $\pi : [0,1] \to \Theta$ between $\theta$ and $\theta_1$ passing through each $\theta_i$ such that $L(f_{\pi(t)}) \le L(f_\theta) + \epsilon$ for $0 \le t \le 1$.*

Though naïvely one might expect to be able to directly connect the weights of $\theta$ and $\theta_1$ via interpolation, such a path may incur high loss as the loss function is not convex over $\Theta$. In our proof of Lemma 1, we rely on a much more careful construction. The construction uses two types of steps: (a) interpolate between two weights in the top layer (the loss is convex in the top layer weights); (b) if a set of neurons already have their output weights set to zero, then we can change their input weights arbitrarily. See Figure 1 for an example path for a 3-layer network. Here we have separated the weight matrices into equally sized blocks: $A_3 = [\ L_3\ |\ R_3\ ]$, $A_2 = \left[\begin{array}{c|c} L_2 & C_2 \\ \hline D_2 & R_2 \end{array}\right]$ and $A_1 = \left[\begin{array}{c} L_1 \\ \hline B_1 \end{array}\right]$. The path consists of 6 steps alternating between type (a) and type (b). Note that for all the type (a) steps, we only update the top layer weights; for all the type (b) steps, we only change rows of a weight matrix (inputs to neurons) if the corresponding columns in the previous matrix (outputs of neurons) are already 0. In Section A we show how such a path can be generalized to any number of layers.

$$
\begin{array}{cccc}
 & A_3 & A_2 & A_1 \\
(1) & [\ L_3\ |\ R_3\ ] & \left[\begin{array}{c|c} L_2 & C_2 \\ \hline D_2 & R_2 \end{array}\right] & \left[\begin{array}{c} L_1 \\ \hline B_1 \end{array}\right] & \\
(2) & [\ rL_3\ |\ 0\ ] & \left[\begin{array}{c|c} L_2 & C_2 \\ \hline D_2 & R_2 \end{array}\right] & \left[\begin{array}{c} L_1 \\ \hline B_1 \end{array}\right] & (a) \\
(3) & [\ rL_3\ |\ 0\ ] & \left[\begin{array}{c|c} L_2 & C_2 \\ \hline rL_2 & 0 \end{array}\right] & \left[\begin{array}{c} L_1 \\ \hline B_1 \end{array}\right] & (b) \\
(4) & [\ 0\ |\ rL_3\ ] & \left[\begin{array}{c|c} L_2 & C_2 \\ \hline rL_2 & 0 \end{array}\right] & \left[\begin{array}{c} L_1 \\ \hline B_1 \end{array}\right] & (a) \\
(5) & [\ 0\ |\ rL_3\ ] & \left[\begin{array}{c|c} rL_2 & 0 \\ \hline rL_2 & 0 \end{array}\right] & \left[\begin{array}{c} L_1 \\ \hline B_1 \end{array}\right] & (b) \\
(6) & [\ rL_3\ |\ 0\ ] & \left[\begin{array}{c|c} rL_2 & 0 \\ \hline rL_2 & 0 \end{array}\right] & \left[\begin{array}{c} L_1 \\ \hline B_1 \end{array}\right] & (a) \\
(7) & [\ rL_3\ |\ 0\ ] & \left[\begin{array}{c|c} rL_2 & 0 \\ \hline 0 & 0 \end{array}\right] & \left[\begin{array}{c} L_1 \\ \hline 0 \end{array}\right] & (b)
\end{array}
$$

Figure 1: Example path, 6 line segments from a 3-layer network to its dropout version. Red denotes weights that have changed between steps while green denotes the zeroed weights that allow us to make these changes without affecting our output.

We then show that we can permute the hidden units of $\theta_1^A$ such that its non-zero units do not intersect with those of $\theta_1^B$, thus allowing us two interpolate between these two parameters. This is formalized in the following lemma and the proof is deferred to supplementary material.

**Lemma 2.** *Let $\theta$ and $\theta'$ be two solutions such that at least $\lceil h_i/2 \rceil$ of the units in the $i^{th}$ hidden layer have been set to zero in both. Then there exists a path in parameter space $\pi : [0,1] \to \Theta$ between $\theta$ and $\theta'$ with 8 line segments such that $L(f_{\pi(t)}) \leq \max\{L(f_\theta), L(f_{\theta'})\}$.*

Theorem 1 follows immediately from Lemma 1 and Lemma 2, as one can first connect $\theta^A$ to its dropout version $\theta_1^A$ using Lemma 1, then connect $\theta_1^A$ to dropout version $\theta_1^B$ of $\theta^B$ using Lemma 2, and finally connect $\theta_1^B$ to $\theta^B$ using Lemma 1 again.

Finally, our results can be generalized to convolutional networks if we do *channel-wise* dropout (Tompson et al., 2015; Keshari et al., 2018).

**Remark 1.** *For convolutional networks, a* channel-wise *dropout will randomly set entire channels to 0 and rescale the remaining channels using an appropriate factor. Theorem 1 can be extended to work with channel-wise dropout on convolutional networks.*

# 4 Connectivity via noise stability

In this section, we relate mode connectivity to another notion of robustness for neural networks—noise stability. It has been observed (Morcos et al., 2018) that neural networks often perform as well even if a small amount of noise is injected into the hidden layers. This was formalized in (Arora et al., 2018), where the authors showed that noise stable networks tend to generalize well. In this section we use a very similar notion of noise stability, and show that all noise stable solutions can be connected as long as the network is sufficiently overparametrized.

We begin in Section 4.1 by restating the definitions of noise stability in (Arora et al., 2018) and also highlighting the key differences in our definitions. In Section 6 we verify these assumptions in practice. In Section 4.2, we first prove that noise stability implies dropout stability (meaning Theorem 1 applies) and then show that it is in fact possible to connect noise stable neural networks via even simpler paths than mere dropout stable networks.

## 4.1 Noise stability

First we introduce some additional notations and assumptions. In this section, we consider a finite and fixed training set $S$. For a network parameter $\theta$, the empirical loss function is $L(\theta) = \frac{1}{|S|}\sum_{(x,y)\in S} l(y, f(x))$. Here the loss function $l(y, \hat{y})$ is assumed to be $\beta$-Lipschitz in $\hat{y}$: for any $\hat{y}, \hat{y}' \in \mathbb{R}^{h_d}$ and any $y \in \mathbb{R}^{h_d}$, we have $|l(y, \hat{y}) - l(y, \hat{y}')| \leq \beta \|\hat{y} - \hat{y}'\|$. Note that the standard cross entropy loss over the softmax function is $\sqrt{2}$-Lipschitz.

For any two layers $i \leq j$, let $M^{i,j}$ be the operator for the composition of these layers, such that $x^j = M^{i,j}(x^i)$. Let $J_{x^i}^{i,j}$ be the Jacobian of $M^{i,j}$ at input $x^i$. Since the activation functions are ReLU's, we know $M^{i,j}(x^i) = J_{x^i}^{i,j} x^i$.

Arora et al. (2018) used several quantities to define noise stability. We state the definitions of these quantities below.

**Definition 2** (Noise Stability Quantities)**.** *Given a sample set $S$, the **layer cushion** of layer $i$ is defined as $\mu_i := \min_{x \in S} \frac{\|A_i \phi(x^{i-1})\|}{\|A_i\|_F \|\phi(x^{i-1})\|}$.*

*For any two layers $i \leq j$, the **interlayer cushion** $\mu_{i,j}$ is defined as $\mu_{i,j} = \min_{x \in S} \frac{\|J_{x^i}^{i,j} x^i\|}{\|J_{x^i}^{i,j}\| \|x^i\|}$.*

*Furthermore, for any layer $i$ the **minimal interlayer cushion** is defined as[2] $\mu_{i\to} = \min_{i \leq j \leq d} \mu_{i,j}$.*

*The **activation contraction** $c$ is defined as $c = \max_{x \in S,\ 1 \leq i \leq d-1} \frac{\|x^i\|}{\|\phi(x^i)\|}$.*

Intuitively, these quantities measures the stability of the network's output to noise for both a single layer and across multiple layers. Note that the definition of the *interlayer cushion* is slightly different

from the original definition in (Arora et al., 2018). Specifically, in the denominator of our definition of interlayer cushion, we replace the Frobenius norm of $J_{x^i}^{i,j}$ by its spectral norm. In the original definition, the interlayer cushion is at most $1/\sqrt{h_i}$, simply because $J_{x^i}^{i,i} = I_{h_i}$ and $\mu_{i,i} = 1/\sqrt{h_i}$. With this new definition, the interlayer cushion need not depend on the layer width $h_i$.

The final quantity of interest is interlayer smoothness, which measures how close the network's behavior is to its linear approximation under noise. Our focus here is on the noise generated by the dropout procedure (Algorithm 1). Let $\theta = \{A_1, A_2, ..., A_d\}$ be weights of the original network, and let $\theta^i = \{A_1, \hat{A}_2, \ldots, \hat{A}_i, A_{i+1}, \ldots, A_d\}$ be the result of applying Algorithm 1 to weight matrices from layer 2 to layer $i$.[3] For any input $x$, let $\hat{x}_i^i(t)$ and $\hat{x}_{i-1}^i(t)$ be the vector before activation at layer $i$ using parameters $\theta t + \theta^i(1-t)$ and $\theta t + \theta^{i-1}(1-t)$ respectively.

**Definition 3** (Interlayer Smoothness). *Given the scenario above, define **interlayer smoothness** $\rho$ to be the largest number such that with probability at least $1/2$ over the randomness in Algorithm 1 for any two layers $i, j$ satisfying for every $2 \leq i \leq j \leq d$, $x \in S$, and $0 \leq t \leq 1$*

$$\|M^{i,j}(\hat{x}_i^i(t)) - J_{x^i}^{i,j}(\hat{x}_i^i(t))\| \leq \frac{\|\hat{x}_i^i(t) - x^i\|\|x^j\|}{\rho\|x^i\|},$$

$$\|M^{i,j}(\hat{x}_{i-1}^i(t)) - J_{x^i}^{i,j}(\hat{x}_{i-1}^i(t))\| \leq \frac{\|\hat{x}_{i-1}^i(t) - x^i\|\|x^j\|}{\rho\|x^i\|}.$$

If the network is smooth (has Lipschitz gradient), then interlayer smoothness holds as long as $\|\hat{x}_i^i(t) - x^i\|, \|\hat{x}_{i-1}^i(t) - x^i\|$ is small. Essentially the assumption here is that the network behaves smoothly in the random directions generated by randomly dropping out columns of the matrices.

Similar to (Arora et al., 2018), we have defined multiple quantities measuring the noise stability of a network. These quantities are in practice small constants as we verify experimentally in Section 6. Finally, we combine all these quantities to define a single overall measure of the noise stability of a network.

**Definition 4** (Noise Stability). *For a network $\theta$ with layer cushion $\mu_i$, minimal interlayer cushion $\mu_{i\rightarrow}$, activation contraction $c$ and interlayer smoothness $\rho$, if the minimum width layer $h_{min}$ is at least $\widetilde{\Omega}(1)$ wide, $\rho \geq 3d$ and $\|\phi(\hat{x}_i^i(t))\|_\infty = O(1/\sqrt{h_i})\|\phi(\hat{x}_i^i(t))\|$ for $1 \leq i \leq d-1, 0 \leq t \leq 1$, we say the network $\theta$ is $\epsilon$-noise stable for*

$$\epsilon = \frac{\beta c d^{3/2} \max_{x \in S}(\|f_\theta(x)\|)}{h_{\min}^{1/2} \min_{2 \leq i \leq d}(\mu_i \mu_{i\rightarrow})}.$$

The smaller $\epsilon$, the more robust the network. Note that the quantity $\epsilon$ is small as long as the hidden layer width $h_{\min}$ is large compared to the noise stable parameters. Intuitively, we can think of $\epsilon$ as a single parameter that captures the noise stability of the network.

## 4.2 Noise stability implies dropout stability

We now show that noise stable local minimizers must also be dropout stable, from which it follows that noise stable local minimizers are connected. We first define the dropout procedure we will be using in Algorithm 1.

The main theorem that we prove in this section is:

**Theorem 2.** *Let $\theta^A$ and $\theta^B$ be two fully connected networks that are both $\epsilon$-noise stable, there exists a path with 10 line segments in parameter space $\pi : [0,1] \rightarrow \Theta$ between $\theta^A$ and $\theta^B$ such that[4] $L(f_{\pi(t)}) \leq \max\{L(f_{\theta^A}), L(f_{\theta^B})\} + \widetilde{O}(\epsilon)$ for $0 \leq t \leq 1$.*

To prove the theorem, we will first show that the networks $\theta^A$ and $\theta^B$ are $\widetilde{O}(\epsilon)$-dropout stable. This is captured in the following main lemma:

**Algorithm 1** Dropout $(A_i, p)$

---

**Input:** Layer matrix $A_i \in \mathbb{R}^{h_i \times h_{i-1}}$, dropout probability $0 < p < 1$.
**Output:** Returns $\hat{A}_i \in \mathbb{R}^{h_i \times h_{i-1}}$.
 1: For each $j \in [h_{i-1}]$, let $\delta_j$ be an i.i.d. Bernoulli random variable which takes the value $0$ with probability $p$ and takes the value $\frac{1}{1-p}$ with probability $(1-p)$.
 2: For each $j \in [h_{i-1}]$, let $[\hat{A}_i]_j$ be $\delta_j[A_i]_j$, where $[\hat{A}_i]_j$ and $[A_i]_j$ are the $j$-th column of $\hat{A}_i$ and $A_i$ respectively.

---

**Lemma 3.** *Let $\theta$ be an $\epsilon$-noise stable network, and let $\theta_1$ be the network with weight matrices from layer 2 to layer $d$ dropped out by Algorithm 1 with dropout probability $\widetilde{\Omega}(1/h_{min}) < p \le \frac{3}{4}$. For any $2 \le i \le d$, assume $\|[A_i]_j\| = O(\sqrt{p})\|A_i\|_F$ for $1 \le j \le h_{i-1}$. For any $0 \le t \le 1$, define the network on the segment from $\theta$ to $\theta_1$ as $\theta_t := \theta + t(\theta_1 - \theta)$. Then, with probability at least $1/4$ over the weights generated by Algorithm 1, $L(f_{\theta_t}) \le L(f_\theta) + \widetilde{O}(\sqrt{p}\epsilon)$, for any $0 \le t \le 1$.*

The main difference between Lemma 3 and Lemma 1 is that we can now directly interpolate between the original network and its dropout version, which reduces the number of segments required. This is mainly because in the noise stable setting, we can prove that after dropping out the neurons, not only does the output remains stable but moreover *every* intermediate layer also remains stable.

From Lemma 3, the proof of Theorem 2 is very similar to the proof of Theorem 1. The detailed proof is given in Section B.

The additional power of Lemma 3 also allows us to consider a smaller dropout probability. The theorem below allows us to trade the dropout fraction with the energy barrier $\epsilon$ that we can prove—if the network is highly overparametrized, one can choose a small dropout probability $p$ which allow the energy barrier $\epsilon$ to be smaller.

**Theorem 3.** *Suppose there exists a network $\theta^*$ with layer width $h_i^*$ for each layer $i$ that achieves loss $L(f_{\theta^*})$, and minimum hidden layer width $h_{min}^* = \widetilde{\Omega}(1)$. Let $\theta^A$ and $\theta^B$ be two $\epsilon$-noise stable networks. For any dropout probability $1.5 \max_{1 \le i \le d-1}(h_i^*/h_i) \le p \le 3/4$, if for any $2 \le i \le d$, $1 \le j \le h_{i-1}$, $\|[A_i]_j\| = O(\sqrt{p})\|A_i\|_F$ then there exists a path with 13 line segments in parameter space $\pi : [0,1] \to \Theta$ between $\theta^A$ and $\theta^B$ such that $L(f_{\pi(t)}) \le \max\{L(f_{\theta^A}) + \widetilde{O}(\sqrt{p}\epsilon), L(f_{\theta^B}) + \widetilde{O}(\sqrt{p}\epsilon), L(f_{\theta^*})\}$ for $0 \le t \le 1$.*

Intuitively, we prove this theorem by connecting $\theta^A$ and $\theta^B$ via the neural network $\theta^*$ with narrow hidden layers. The detailed proof is given in Section B.

## 5   Disconnected modes in two-layer nets

The mode connectivity property is not true for every neural network. Freeman and Bruna (2016) gave a counter-example showing that if the network is not overparametrized, then there can be different global minima of the neural network that are not connected. Venturi et al. (2018) showed that spurious valleys can exist for 2-layer ReLU nets with an arbitrary number of hidden units, but again they do not extend their result to the overparametrized setting. In this section, we show that even if a neural network is overparametrized—in the sense that there exists a network of smaller width that can achieve optimal loss—there can still be two global minimizers that are not connected.

In particular, suppose we are training a two-layer ReLU student network with $h$ hidden units to fit a dataset generated by a ground truth two-layer ReLU teacher network with $h_t$ hidden units such that the samples in the dataset are drawn from some input distribution and the labels computed via forward passes through the teacher network. The following theorem demonstrates that regardless of the degree to which the student network is overparametrized, we can always construct such a dataset for which global minima are not connected.

**Theorem 4.** *For any width $h$ and and convex loss function $l : \mathbb{R} \times \mathbb{R} \mapsto \mathbb{R}$ such that $l(y, \hat{y})$ is minimized when $y = \hat{y}$, there exists a dataset generated by ground-truth teacher network with two hidden units (i.e. $h_t = 2$) and one output unit such that global minimizers are not connected for a student network with $h$ hidden units.*

Our proof is based on an explicit construction. The detailed construction is given in Section C.

## 6 Experiments

We now demonstrate that our assumptions and theoretical findings accurately characterize mode connectivity in practical settings. In particular, we empirically validate our claims using standard convolutional architectures—for which we treat individual filters as the hidden units and apply channel-wise dropout (see Remark 1)—trained on datasets such as CIFAR-10 and MNIST.

Training with dropout is not necessary for a network to be either dropout-stable or noise-stable. Recall that our definition of dropout-stability merely requires the existence of a particular sub-network with half the width of the original that achieves low loss. Moreover, as Theorem 3 suggests, if there exists a narrow network that achieves low loss, then we need only be able to drop out a number of filters equal to the width of the narrow network to connect local minima.

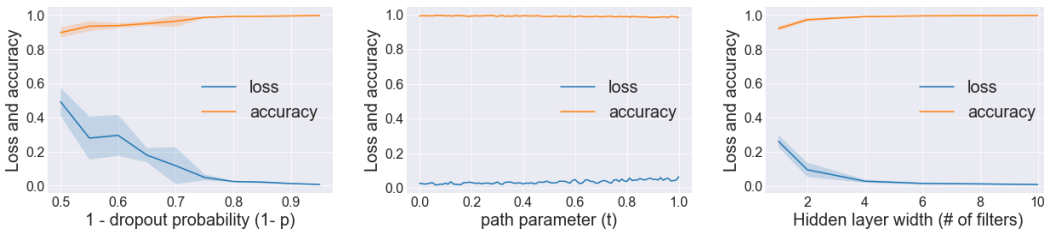

Figure 2: Results for convolutional networks trained on MNIST.

First, we demonstrate in the left plot in Figure 2 on MNIST that 3-layer convolutional nets (not counting the output layer) with 32 $3 \times 3$ filters in each layer tend to be fairly dropout stable—both in the original sense of Definition 1 and especially if we relax the definition to allow for wider subnetworks—despite the fact that no dropout was applied in training. For each trial, we randomly sampled 20 dropout networks with *exactly* $\lfloor 32(1-p) \rfloor$ non-zero filters in each layer and report the performance of the best one. In the center plot, we verify for $p = 0.2$ we can construct a linear path $\pi(t) : \mathbb{R} \rightarrow \Theta$ from our convolutional net to a dropout version of itself. Similar results were observed when varying $p$. Finally, in the right plot we demonstrate the existence of 3-layer convolutional nets just a few filters wide that are able to achieve low loss on MNIST. Taken together, these results indicate that our path construction in Theorem 3 performs well in practical settings. In particular, we can connect two convolutional nets trained on MNIST by way of first interpolating between the original nets and their dropped out versions with $p = 0.2$, and then connecting the dropped out versions by way of a narrow subnetwork with at most $\lfloor 32p \rfloor$ non-zero filters.

We also demonstrate that the VGG-11 (Simonyan and Zisserman, 2014) architecture trained with channel-wise dropout (Tompson et al., 2015; Keshari et al., 2018) with $p = 0.25$ at the first three layers[5] and $p = 0.5$ at the others on CIFAR-10 converges to a noise stable minima—as measured by layer cushion, interlayer cushion, activation contraction and interlayer smoothness. The network under investigation achieves 95% training and 91% test accuracy with channel-wise dropout *activated*, in comparison to 99% training and 92% test accuracy with dropout turned off. Figure 3 plots the distribution of the noise stability parameters over different data points in the training set, from which we can see they behave nicely. Interestingly, we also discovered that networks trained without channel-wise dropout exhibit similarly nice behavior on all but the first few layers. Finally, in Figure 3, we demonstrate that the training loss and accuracy obtained via the path construction in Theorem 3 between two noise stable VGG-11 networks $\theta_A$ and $\theta_B$ remain fairly low and high respectively—particularly in comparison to directly interpolating between the two networks, which incurs loss as high as 2.34 and accuracy as low as 10%, as shown in Section D.2.

Further details on all experiments are provided in Section D.1.

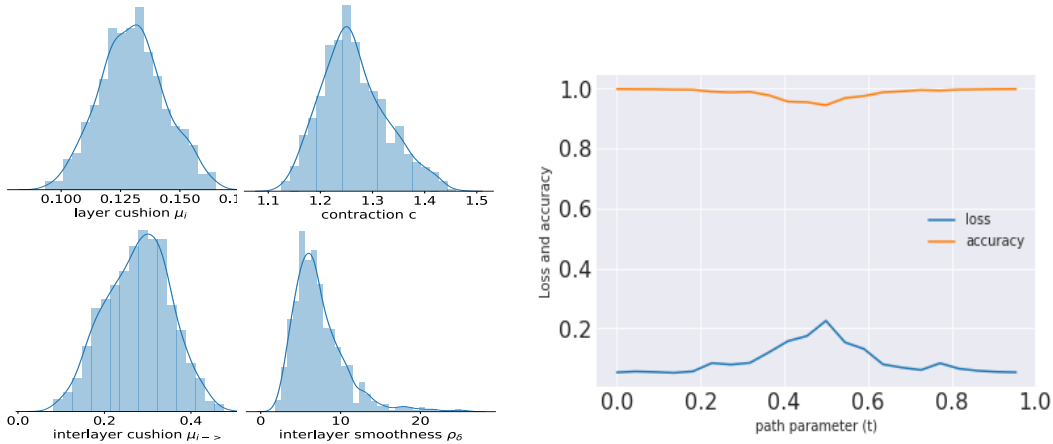

Figure 3: Left) Distribution of layer cushion, activation contraction, interlayer cushion and interlayer smoothness of the 6-th layer of a VGG-11 network on the training set. The other layers' parameters are exhibited in Section D.3. Right) The loss and training accuracy along the path between two noise stable VGG-11 networks described in Theorem 3.

**Acknowledgments**

Rong Ge acknowledges funding from NSF CCF-1704656, NSF CCF-1845171 (CAREER), the Sloan Fellowship and Google Faculty Research Award. Sanjeev Arora acknowledges funding from the NSF, ONR, Simons Foundation, Schmidt Foundation, Amazon Research, DARPA and SRC.

## Footnotes

[1] Note our results will also work if $r$ is allowed to vary for each layer.

[2]Note that $J_{x^i}^{i,i} = I_{h_i}$ and $\mu_{i,i} = 1$.

[3]Note that $A_1$ is excluded because dropping out columns in $\hat{A}_2$ already drops out the neurons in layer 1; dropping out columns in $A_1$ would drop out input coordinates, which is not necessary.

[4]Here $\widetilde{O}(\cdot)$ hides log factors on relevant factors including $|S|, d, \|x\|, 1/\epsilon$ and $h_i\|A_i\|$ for layers $i \in [d]$.

[5]we find the first three layers are less resistant to channel-wise dropout.

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
