[Supplementary Material · camera_ready_full.pdf]

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

# A  Proofs for connectivity of dropout-stable optima

**Proof of Lemma 1.** Without loss of generality, suppose for each $\theta_i$ that the subset of $\lfloor h_i/2 \rfloor$ non-zero hidden units in each layer are all indexed between 1 and $\lfloor h_i/2 \rfloor$. For $1 < i < d$, we can partition $A_i$ into quadrants such that $A_i = \left[ \begin{array}{c|c} L_i & C_i \\ \hline D_i & R_i \end{array} \right]$. (Here, $L_i \in \mathbb{R}^{\lfloor h_i/2 \rfloor \times \lfloor h_i/2 \rfloor}$. If $h_i$ is odd, when we write $L_i$ in the other quadrants we implicitly pad it with zeros in a consistent manner.) Similarly, we can partition $A_1$ such that $A_1 = \left[ \begin{array}{c} L_1 \\ \hline B_1 \end{array} \right]$ and $A_d$ such that $A_d = \left[ \begin{array}{c|c} L_d & R_d \end{array} \right]$. We will sometimes use the notation $A_i$ to refer to the value of $A_i$ at a given point on our path, while $A_i^\theta$ will always refer to the value of $A_i$ at $\theta$. We now proceed to prove via induction the existence of a path from $\theta$ to $\theta_i$ for all $i$ whose loss is bounded by $L(f_\theta) + \epsilon$, from which the main result immediately follows.

**Base case: from $\theta$ to $\theta_{d-1}$** As a base case of the induction, we need to construct a path from $\theta$ to $\theta_{d-1}$, such that the loss is bounded by $L(f_\theta) + \epsilon$. First, note that setting a particular subset of columns (e.g. the right half of columns) in $A_i$ to zero is equivalent to setting the corresponding rows (e.g. the bottom half of rows) of $A_{i-1}$ to zero. So from the fact that $L(f_{\theta_{d-1}}) \le L(f_\theta) + \epsilon$ it follows that we can equivalently replace $A_d^\theta$ with $\left[ \begin{array}{c|c} rL_d^\theta & 0 \end{array} \right]$ without increasing our loss by more than $\epsilon$.

In fact, because our loss function is convex over $A_d$ we can actually interpolate $A_d$ between $A_d^\theta$ and $\left[ \begin{array}{c|c} rL_d^\theta & 0 \end{array} \right]$ while keeping our loss below $L(f_\theta) + \epsilon$ at every point along this subpath.

Then, because $R_d = 0$ we can modify both $D_{d-1}$ and $R_{d-1}$ any way we'd like without affecting the output of our network. In particular, we can interpolate $A_{d-1}$ between $A_{d-1}^\theta$ and $\left[ \begin{array}{c|c} L_{d-1}^\theta & C_{d-1}^\theta \\ \hline 0 & 0 \end{array} \right]$ while keeping our loss constant long this subpath, thus arriving at $\theta_{d-1}$.

**From $\theta_k$ to $\theta_{k-1}$**

Suppose we have found a path from $\theta$ to $\theta_k$ such that (1) $A_d^{\theta_k} = \left[ \begin{array}{c|c} rL_d^\theta & 0 \end{array} \right]$, (2) $A_i^{\theta_k} = \left[ \begin{array}{c|c} rL_i^\theta & 0 \\ \hline 0 & 0 \end{array} \right]$ for $k < i < d$, (3) $A_k^{\theta_k} = \left[ \begin{array}{c|c} L_k^\theta & C_k^\theta \\ \hline 0 & 0 \end{array} \right]$, and (4) $A_i^{\theta_k} = A_i^\theta$ for $i < k$, such that the loss along the path is at most $L(f_\theta) + \epsilon$. Note that $\theta_{d-1}$ satisfies all these assumptions, including in particular (2) as there are of course no $A_i$ between $A_{d-1}$ and $A_d$. Now let us extend this path to $\theta_{k-1}$.

First, because the rightmost columns of $A_i$ are zero for $k < i \le d$, we can modify the bottom rows of $A_i$ for $k \le i < d$ without affecting the output of our network. In particular, we can set $A_k$ to $\left[ \begin{array}{c|c} L_k^\theta & C_k^\theta \\ \hline rL_k^\theta & 0 \end{array} \right]$, as well as $A_i$ to $\left[ \begin{array}{c|c} rL_i^\theta & 0 \\ \hline 0 & rL_i^\theta \end{array} \right]$ for $k < i < d$. From the fact that the loss is convex over $A_d$ and that $L(f_{\theta_{k-1}}) < L(f_\theta) + \epsilon$, it then follows that we can set $A_d$ to $\left[ \begin{array}{c|c} 0 & rL_d^\theta \end{array} \right]$ via interpolation while keeping our loss below $L(f_\theta) + \epsilon$. In particular, note that because the off-diagonal blocks of $A_i$ are zero for $k < i < d$, interpolating between the leftmost columns of $A_d$ being non-zero and the rightmost columns of $A_d$ being non-zero simply amounts to interpolating between the outputs of the two subnetworks comprised respectively of the first $\lfloor h_i/2 \rfloor$ and last $\lfloor h_i/2 \rfloor$ rows of $A_i$ for $k \le i < d$.

Once we have the leftmost columns of $A_d$ set to zero and $A_i$ in block-diagonal form for $k < i < d$, we can proceed to modify the top rows of $A_k$ however we'd like without affecting the output of our network. Specifically, let us set $A_k$ to $\left[ \begin{array}{c|c} rL_k^\theta & 0 \\ \hline rL_k^\theta & 0 \end{array} \right]$. We can then reset $A_d$ to $\left[ \begin{array}{c|c} rL_d^\theta & 0 \end{array} \right]$ via interpolation—this time without affecting our loss since the weights of our two subnetworks are equivalent—and afterwards set $D_k$ to zero and $R_i$ to zero for $k \le i < d$—again without affecting our loss since the rightmost columns of $A_d$ are now zero, meaning that the bottom rows of $A_i$ have no affect on our network's output.

Following these steps, we will have $A_i = \left[\begin{array}{c|c} rL_i^\theta & 0 \\ \hline 0 & 0 \end{array}\right]$ for $k \leq i < d$ and $A_d = \left[\begin{array}{c|c} rL_d^\theta & 0 \end{array}\right]$. And so we are now free to set the bottom rows of $A_{k-1}$ to zero without affecting our loss, thus arriving at $\theta_{k-1}$. $\qquad\square$

**Lemma 4.** *Let $\theta$ be a parameter such that at least $\lceil h_i/2 \rceil$ of the units in each hidden layer have been set to zero. Then we can achieve an arbitrary permutation of the non-zero hidden units of $\theta$ via a path consisting of just 5 line segments such that our loss is constant along this path.*

*Proof.* Let $\pi : [h_i] \mapsto [h_i]$ be some permutation over the units in layer $i$. Without loss of generality, suppose all non-zero units in layer $i$ are indexed between 0 and $\lfloor h_i/2 \rfloor$, and define $\pi' : [\lfloor h_i/2 \rfloor] \mapsto [h_i] \setminus [\lfloor h_i/2 \rfloor]$ as any one-to-one mapping such that $\pi'(i) = \pi(i)$ if $\pi(i) \in [h_i] \setminus [\lfloor h_i/2 \rfloor]$. Note that when we refer to a unit $j$ as "set to zero", we mean that both row $j$ of $A_i$ and column $j$ of $A_{i+1}$ have been set to zero.

To permute the units of layer $i$, we can first simultaneously copy the non-zero rows of $A_i$ into a subset of the rows that have been set to zero. Specifically, for $j \in [\lfloor h_i/2 \rfloor]$ we can copy row $j$ of $A_i$ into row $\pi'(j)$ via interpolation and without affecting our loss, due to the fact that column $\pi'(j)$ in $A_{i+1}$ is set to zero. We can then set column $j$ of $A_{i+1}$ to zero while copying its value to column $\pi'(j)$, again via interpolation and without affecting our loss since rows $j$ and $\pi'(j)$ of $A_i$ are now equivalent.

Following these first two steps, the first $\lfloor h_i/2 \rfloor$ columns of $A_{i+1}$ will have been set to zero. Thus, for all $j \in [\lfloor h_i/2 \rfloor]$ such that $\pi(j) \in [h_i/2]$ we can copy row $\pi'(j)$ of $A_i$ into row $\pi(j)$ without affecting our loss. We can then set column $\pi'(j)$ of $A_{i+1}$ to zero while copying its value into column $\pi(j)$ via interpolation and without affecting our loss since rows $\pi'(j)$ and $\pi(j)$ of $A_i$ are now equivalent. Setting row $\pi'(j)$ to zero—again for all $j \in [\lfloor h_i/2 \rfloor]$ such that $\pi(j) \in [h_i/2]$—completes the permutation for layer $i$.

Note that because we leave the output of layer $i$ unchanged throughout the course of permuting the units of layer $i$, it follows that we can perform all swaps across all layers simultaneously. And so from the fact that permuting each layer can be done in 5 steps—each of which consists of a single line segment in parameter space—the main result immediately follows. $\qquad\square$

**Proof of Lemma 2.** Without loss of generality, suppose for $\theta$ that the subset of $\lfloor h_i/2 \rfloor$ non-zero hidden units in each layer $i$ are all indexed between 0 and $\lfloor h_i/2 \rfloor$. Note that when we refer to a unit as "set to zero", we mean that both the corresponding row of $A_i$ and column of $A_{i+1}$ have been set to zero. Adopting our notation in Lemma 1, we can construct a path from $\theta$ to $\theta'$ as follows.

First, from the fact that the second half of units in each hidden layer $i$ have been set to zero in $\theta$ we have that $A_1^\theta = \left[\begin{array}{c} L_1^\theta \\ \hline 0 \end{array}\right]$, $A_i^\theta = \left[\begin{array}{c|c} L_i^\theta & 0 \\ \hline 0 & 0 \end{array}\right]$ for $1 < i < d$, and $A_d^\theta = \left[\begin{array}{c|c} L_d^\theta & 0 \end{array}\right]$. Similarly, half the rows of $A_1^{\theta'}$ are zero, half the rows and columns of $A_i^{\theta'}$ are zero for $1 < i < d$, and half the columns of $A_d^{\theta'}$ are zero. Note that the indices of the non-zero units in $\theta'$ may intersect with those of the non-zero units in $\theta$. For $1 \leq i \leq d$, let $B_i$ denote the submatrix of $A_i$ corresponding to the non-zero rows and columns of $A_i^{\theta'}$.

Because $A_i^\theta$ are block-diagonal for $1 < i < d$ and the rightmost columns of $A_d^\theta$ are zero, starting from $\theta$ we can modify the bottom rows of $A_i$ for $1 \leq i < d$ any way we'd like without affecting our loss—as done in our path construction for Lemma 1. In particular, let us set $A_i$ to $\left[\begin{array}{c|c} L_i^\theta & 0 \\ \hline 0 & B_i^{\theta'} \end{array}\right]$ for $1 < i < d$ and $A_1$ to $\left[\begin{array}{c} L_1^\theta \\ \hline B_1^{\theta'} \end{array}\right]$. Then, from the fact that our loss function is convex over $A_d$ it follows that we can set $A_d$ to $\left[\begin{array}{c|c} 0 & B_d^{\theta'} \end{array}\right]$ via interpolation while keeping our loss below $\max\{L(f_\theta), L(f_{\theta'})\}$. Finally, from the fact that the leftmost columns of $A_d$ are now zero and $A_i$ are still block-diagonal for $1 < i < d$, it follows that we can set $L_i$ to zero for $1 \leq i < d$ without affecting our loss—thus making $A_i$ equal to $\left[\begin{array}{c|c} 0 & 0 \\ \hline 0 & B_i^{\theta'} \end{array}\right]$ for $1 < i < d$ and $A_1$ equal to $\left[\begin{array}{c} 0 \\ \hline B_1^{\theta'} \end{array}\right]$.

To complete our path from $\theta$ to $\theta'$ we now simply need to permute the units of each hidden layer so as to return the elements of $B_i^{\theta'}$ to their original positions in $A_i$ for each $i$. From Lemma 4 it follows

that we can accomplish this permutation via 5 line segments in parameter space without affecting our loss. Combined with the previous steps above, we have constructed path from $\theta$ to $\theta'$ consisting of a total of 8 line segments whose loss is bounded by $\max\{L(f_\theta), L(f_{\theta'})\}$. $\qquad\square$

**Proof of Theorem 1.** First, from Lemma 1 we know we can construct paths from both $\theta^A$ to $\theta_1^A$ and $\theta^B$ to $\theta_1^B$ while keeping our loss below $L(f_{\theta^A}) + \epsilon$ and $L(f_{\theta^B}) + \epsilon$ respectively. From Lemma 2 we know that we can construct a path from $\theta_1^A$ to $\theta_1^B$ such that the loss along the path is bounded by $\max\{L(f_{\theta_1^A}), L(f_{\theta_1^B})\}$. The main result then follows from the fact that $L(f_{\theta_1^A}) \leq L(f_{\theta^A}) + \epsilon$ and $L(f_{\theta_1^B}) \leq L(f_{\theta^B}) + \epsilon$ due to $\theta^A$ and $\theta^B$ both being $\epsilon$-dropout stable. $\qquad\square$

# B  Proofs for connectivity via noise stability

In this section, we give detailed proofs showing that noise stability implies connectivity. In the following lemma, we first show that the network output is stable if we randomly dropout columns in a single layer using Algorithm 1.

**Lemma 5.** *For any layer $2 \leq i \leq d$, let $G = \{(U^{(l)}, x^{(l)})\}_{l=1}^m$ be a set of matrix/vector pairs of size $m$ where $U \in \mathbb{R}^{h_d \times h_i}$ and $x \in \mathbb{R}^{h_{i-1}}$ satisfying $\|x\|_\infty = O\left(\frac{\|x\|}{\sqrt{h_{i-1}}}\right)$. Given $A_i$, let $\hat{A}_i \in \mathbb{R}^{h_i \times h_{i-1}}$ be the output of Algorithm 1 with dropout probability $0 < p \leq \frac{3}{4}$. Assume $\|[A_i]_j\| = O(\sqrt{p})\|A_i\|_F$ for $1 \leq j \leq h_{i-1}$. Given any $0 < \delta < 1$, let $\epsilon' = O\left(\sqrt{\frac{\log(mh_d/\delta)p}{h_{\min}}}\right)$, with probability at least $1 - \delta$, we have for any $(U, x) \in G$ that $\|U(\hat{A}_i - A_i)x\| \leq \epsilon' \|A_i\|_F \|U\|\|x\|$. Further assuming $h_{\min} = \Omega\left(\frac{\log(1/\delta)}{p}\right)$, we know with probability at least $1 - \delta$, no less than $\frac{2}{3}p$ fraction of columns in $\hat{A}_i$ are zero vectors.*

Intuitively, this lemma upper-bounds the change in the network output after dropping out a single layer. In the lemma, we should think of $x$ as the input to the current layer, $A_i$ as the layer matrix and $U$ as the Jacobian of the network output with respect to the layer output. If the activation pattern does not change after the dropping out, $U\hat{A}_i x$ is exactly the output of the dropped out network and $\|U(\hat{A}_i - A_i)x\|$ is the change in the network output.

**Proof of Lemma 5.** Fixing $2 \leq i \leq d$ and one pair $(U, x) \in G$, we show with probability at least $1 - \frac{\delta}{m}$, $\|U(\hat{A}_i - A_i)x\| \leq \epsilon' \|A_i\|_F \|U\|\|x\|$. Let $U_k$ be the $k$-th column of $U$. Then by definition of $\hat{A}_i$ in the algorithm, we know

$$U(\hat{A}_i - A_i)x = \sum_{k,j} U_k [A_i]_{kj} x_j (\delta_j - 1)$$

$$= \sum_j \left(\sum_k U_k [A_i]_{kj}\right) x_j (\delta_j - 1),$$

where $\delta_j$ is an i.i.d. Bernoulli random variable which takes the value 0 with probability $p$ and takes the value $\frac{1}{1-p}$ with probability $(1-p)$.

Let $[A_i]_j$ be the $j$-th column of $A_i$. Because $p \leq \frac{3}{4}$, $\frac{1}{1-p} = O(1)$ (any $p$ bounded away from 1 will work). Hence the norm for each individual term can be bounded as follows.

$$\left\|\left(\sum_k U_k [A_i]_{kj}\right) x_j (\delta_j - 1)\right\| \overset{(*)}{\leq} O\left(\frac{\|x\|}{\sqrt{h_{i-1}}}\right) \|U[A_i]_j\|$$

$$\leq O\left(\frac{\|x\|}{\sqrt{h_{\min}}}\right) \|U\|\|[A_i]_j\|$$

$$\overset{(\dagger)}{\leq} O\left(\frac{\sqrt{p}\|U\|\|A_i\|_F\|x\|}{\sqrt{h_{\min}}}\right),$$

where (*) uses the assumption that $\|x\|_\infty = O\left(\frac{\|x\|}{\sqrt{h_{i-1}}}\right)$ and (†) holds because we assume $\|[A_i]_j\| = O(\sqrt{p})\|A_i\|_F$ for $1 \le j \le h_{i-1}$.

For the total variance, we have

$$
\begin{aligned}
\sigma^2 &:= \sum_j \mathbb{E}\left[\left\|\left(\sum_k U_k[A_i]_{kj}\right)x_j(\delta_j - 1)\right\|^2\right] \\
&\le \sum_j \|U[A_i]_j\|^2 |x_j|^2 \left((0-1)^2 \times p + \left(\frac{1}{1-p} - 1\right)^2 \times (1-p)\right) \\
&\stackrel{(*)}{=} \sum_j \|U[A_i]_j\|^2 \cdot O\left(\frac{\|x\|^2}{h_{i-1}}\right) \cdot p\left(1 + \frac{p}{1-p}\right) \\
&\le \|UA_i\|_F^2 \cdot O\left(\frac{\|x\|^2}{h_{\min}}\right) \cdot p \\
&\le O\left(\frac{p\|U\|^2\|A_i\|_F^2\|x\|^2}{h_{\min}}\right),
\end{aligned}
$$

where inequality $(*)$ uses the assumption that $\|x\|_\infty = O\left(\frac{\|x\|}{\sqrt{h_{i-1}}}\right)$. Then, by the vector Bernstein inequality (Lemma 8), we know given $0 < \delta < 1$, there exists $\epsilon' = O\left(\sqrt{\frac{p\log(mh_d/\delta)}{h_{\min}}}\right)$, with probability at least $1 - \frac{\delta}{m}$, we have

$$\|U(\hat{A}_i - A_i)x\| \le \epsilon'\|A_i\|_F\|U\|\|x\|.$$

Taking the union bound over all $(U, x)$ pairs in $G$, we know that with probability at least $1 - \delta$, for any $(U, x) \in G$, $\|U(\hat{A}_i - A_i)x\| \le \epsilon'\|A_i\|_F\|U\|\|x\|$.

Suppose $h_{\min} = \Omega\left(\frac{\log(1/\delta)}{p}\right)$; then by the Chernoff bound, we know with probability at least $1 - \delta$, the dropped out fraction is at least $\frac{2}{3}p$. Taking another union bound concludes our proof. $\qquad\square$

Now we are ready to prove Lemma 3. The idea is similar to (Arora et al., 2018), but we give the proof here for completeness.

**Lemma 3.** *Let $\theta$ be an $\epsilon$-noise stable network, and let $\theta_1$ be the network with weight matrices from layer 2 to layer $d$ dropped out by Algorithm 1 with dropout probability $\widetilde{\Omega}(1/h_{min}) < p \le \frac{3}{4}$. For any $2 \le i \le d$, assume $\|[A_i]_j\| = O(\sqrt{p})\|A_i\|_F$ for $1 \le j \le h_{i-1}$. For any $0 \le t \le 1$, define the network on the segment from $\theta$ to $\theta_1$ as $\theta_t := \theta + t(\theta_1 - \theta)$. Then, with probability at least $1/4$ over the weights generated by Algorithm 1, $L(f_{\theta_t}) \le L(f_\theta) + \widetilde{O}(\sqrt{p}\epsilon)$, for any $0 \le t \le 1$.*

**Proof of Lemma 3.** We first bound the difference between the dropped out network $\theta_1$ and the original network $\theta$.

**Bounding** $\|f_\theta(x) - f_{\theta_1}(x)\|$: We first show that with probability at least $1/2 - \delta$, $\|f_\theta(x) - f_{\theta_1}(x)\| = \|x^d - \hat{x}_d^d\| \le \epsilon'\|f_\theta(x)\|$, where $\epsilon'$ will be specified later. For any layer $i \ge 1$ and letting $\hat{x}_i^j$ be the vector before activation at layer $j$ if the weights $A_2, \ldots, A_i$ are replaced by $\hat{A}_2, \ldots, \hat{A}_i$.

According to Lemma 5, for any layer $2 \le i \le d$, given $0 < \delta < 1$, let $\epsilon' = O\left(\sqrt{\frac{pc^2d^2\log(mdh_d/\delta)}{h_{\min}\min_{2\le i\le d}(\mu_i^2\mu_{i\to}^2)}}\right)$, with probability at least $1 - \delta/d$ over $\hat{A}_i$, we have

$$\|U(\hat{A}_i - A_i)x\| \le \frac{\epsilon'\mu_i\mu_{i\to}}{6cd}\|A\|_F\|U\|\|x\| \tag{1}$$

for any $(U, x) \in \{(J_{x^i}^{i,j}, \phi(\hat{x}_{i-1}^{i-1}))|x \in S, i \le j \le d\}$. By taking a union bound over $i$, we know inequality (1) holds with probability at least $1 - \delta$ for every $i$. Recall that the interlayer smoothness

holds with probability at least $1/2$. Taking another union, we know with probability at least $1/2 - \delta$, interlayer smoothness holds and inequality (1) holds for every $2 \leq i \leq d$. Next, conditioning on the success of these two events, we will inductively prove for any $1 \leq i \leq d$, for any $i \leq j \leq d$,

$$\|\hat{x}_i^j - x^j\| \leq (i/d)\epsilon'\|x^j\|.$$

For the base case $i = 1$, since we are not dropping out any weight matrix, the inequality is trivial. For any $1 \leq i - 1 \leq d - 1$, suppose $\|\hat{x}_{i-1}^j - x^j\| \leq \frac{i-1}{d}\epsilon'\|x^j\|$ for any $i - 1 \leq j \leq d$; we prove the induction hypothesis holds for layer $i$.

For any $i \leq j \leq d$ we have

$$\|\hat{x}_i^j - x^j\| = \|(\hat{x}_i^j - \hat{x}_{i-1}^j) + (\hat{x}_{i-1}^j - x^j)\| \leq \|\hat{x}_i^j - \hat{x}_{i-1}^j\| + \|\hat{x}_{i-1}^j - x^j\|.$$

By the induction hypothesis, we know the second term can be bounded by $(i-1)\epsilon'\|x^j\|/d$. Therefore, in order to complete the induction step, it suffices to show that the first term is bounded by $\epsilon'\|x^j\|/d$. For simplicity, we also denote $\hat{x}_{i-1}^{i-1}$ as $\hat{x}^{i-1}$. Let $\Delta_i = \hat{A}_i - A_i$. We can decompose the error into two terms:

$$
\begin{aligned}
\|\hat{x}_i^j - \hat{x}_{i-1}^j\| &= \|M^{i,j}(\hat{A}_i\phi(\hat{x}^{i-1})) - M^{i,j}(A_i\phi(\hat{x}^{i-1}))\| \\
&= \|M^{i,j}(\hat{A}_i\phi(\hat{x}^{i-1})) - M^{i,j}(A_i\phi(\hat{x}^{i-1})) + J_{x^i}^{i,j}(\Delta_i\phi(\hat{x}^{i-1})) - J_{x^i}^{i,j}(\Delta_i\phi(\hat{x}^{i-1}))\| \\
&\leq \|J_{x^i}^{i,j}(\Delta_i\phi(\hat{x}^{i-1}))\| + \|M^{i,j}(\hat{A}_i\phi(\hat{x}^{i-1})) - M^{i,j}(A_i\phi(\hat{x}^{i-1})) - J_{x^i}^{i,j}(\Delta_i\phi(\hat{x}^{i-1}))\|
\end{aligned}
\tag{2}
$$

The first term of (2) can be bounded as follows:

$$
\begin{aligned}
&\|J_{x^i}^{i,j}\Delta_i\phi(\hat{x}^{i-1})\| \\
&\leq (\epsilon'\mu_i\mu_{i\to}/6cd)\|J_{x^i}^{i,j}\|\|A_i\|_F\|\phi(\hat{x}^{i-1})\| \quad &&\text{Lemma 5} \\
&\leq (\epsilon'\mu_i\mu_{i\to}/6cd)\|J_{x^i}^{i,j}\|\|A_i\|_F\|\hat{x}^{i-1}\| \quad &&\phi \text{ (ReLU) is 1-Lipschitz} \\
&\leq (\epsilon'\mu_i\mu_{i\to}/3cd)\|J_{x^i}^{i,j}\|\|A_i\|_F\|x^{i-1}\| \quad &&\text{Induction hypothesis,} \\
& &&\left\|\hat{x}^{i-1} - x^{i-1}\right\| \leq \frac{(i-1)\epsilon'\left\|x^{i-1}\right\|}{d} < \left\|x^{i-1}\right\| \\
&\leq (\epsilon'\mu_i\mu_{i\to}/3d)\|J_{x^i}^{i,j}\|\|A_i\|_F\|\phi(x^{i-1})\| \quad &&\text{Activation Contraction} \\
&\leq (\epsilon'\mu_{i\to}/3d)\|J_{x^i}^{i,j}\|\|A_i\phi(x^{i-1})\| \quad &&\text{Layer Cushion} \\
&= (\epsilon'\mu_{i\to}/3d)\|J_{x^i}^{i,j}\|\|x^i\| \quad &&x^i = A_i\phi(x^{i-1}) \\
&\leq (\epsilon'/3d)\|x^j\| \quad &&\text{Interlayer Cushion} \tag{3}
\end{aligned}
$$

The second term of (2) can be bounded as:

$$
\begin{aligned}
&\|M^{i,j}(\hat{A}_i\phi(\hat{x}^{i-1})) - M^{i,j}(A_i\phi(\hat{x}^{i-1})) - J_{x^i}^{i,j}(\Delta_i\phi(\hat{x}^{i-1}))\| \\
&= \|(M^{i,j} - J_{x^i}^{i,j})(\hat{A}_i\phi(\hat{x}^{i-1})) - (M^{i,j} - J_{x^i}^{i,j})(A_i\phi(\hat{x}^{i-1}))\| \\
&\leq \|(M^{i,j} - J_{x^i}^{i,j})(\hat{A}_i\phi(\hat{x}^{i-1}))\| + \|(M^{i,j} - J_{x^i}^{i,j})(A_i\phi(\hat{x}^{i-1}))\|.
\end{aligned}
\tag{4}
$$

Both terms of (4) can be bounded using the interlayer smoothness condition. For the second term of (4), notice that $A_i\phi(\hat{x}^{i-1}) = \hat{x}_{i-1}^i$. Thus by the induction hypothesis, we know

$$\|A_i\phi(\hat{x}^{i-1}) - x^i\| = \|\hat{x}_{i-1}^i - x^i\| \leq (i-1)\epsilon'\|x^i\|/d \leq \epsilon'\|x^i\|. \tag{5}$$

Now, by interlayer smoothness,

$$
\begin{aligned}
\|(M^{i,j} - J_{x^i}^{i,j})(A_i\phi(\hat{x}^{i-1}))\| &= \|(M^{i,j} - J_{x^i}^{i,j})(x^i + (A_i\phi(\hat{x}^{i-1}) - x^i))\| \\
&\leq \frac{\|A_i\phi(\hat{x}^{i-1}) - x^i\|\|x^j\|}{\rho\|x^i\|} \\
&\overset{(*)}{\leq} \frac{\epsilon'\|x^i\|\|x^j\|}{3d\|x^i\|} = \frac{\epsilon'\|x^j\|}{3d} \tag{6}
\end{aligned}
$$

where in (*) we use (5) and the assumption $\rho \geq 3d$. For the first term of (4), we know $\hat{A}_i\phi(\hat{x}^{i-1}) = \hat{x}^i_{i-1} + \Delta_i\phi(\hat{x}^{i-1})$. Therefore by the induction hypothesis and (3) for $i = j$,

$$\|\hat{A}_i\phi(\hat{x}^{i-1}) - x^i\| \leq \|\hat{x}^i_{i-1} - x^i\| + \|\Delta_i\phi(\hat{x}^{i-1})\| \leq (i-1)\epsilon'\|x^i\|/d + \epsilon'\|x^i\|/3d \leq \epsilon'\|x^i\|,$$

so again we have

$$\|(M^{i,j} - J^{i,j}_{x^i})(\hat{A}_i\phi(\hat{x}^{i-1}))\| \leq (\epsilon'/3d)\|x^j\|. \tag{7}$$

Together, (7) and (6) show that (4) is $\leq \frac{2\epsilon'}{3d}\|x^j\|$. Together with (3) we obtain from 2 that $\|\hat{x}^j_i - \hat{x}^j_{i-1}\| \leq \frac{\epsilon'}{d}\|x^j\|$, and hence that $\|\hat{x}^j_i - x^j\| \leq \frac{i\epsilon'\|x^j\|}{d}$, completing the induction step.

Conditioning on the success of interlayer smoothness and inequality (1), we've shown,

$$\|\hat{x}^j_i - x^j\| \leq (i/d)\epsilon'\|x^j\|,$$

for any $i \leq j \leq d$. Recall that with probability at least $1/2 - \delta$, interlayer smoothness holds and inequality (1) holds for every $2 \leq i \leq d$. Thus, let $\epsilon' = O\left(\sqrt{\frac{pc^2 d^2 \log(mdh_d/\delta)}{h_{\min} \min_{2 \leq i \leq d} \mu_i^2 \mu_{i\rightarrow}^2}}\right)$, we know with probability at least $1/2 - \delta$,

$$\|f_\theta(x) - f_{\theta_1}(x)\| = \|x^d - \hat{x}^d_d\| \leq \epsilon'\|f_\theta(x)\|.$$

**Bounding $\|f_\theta(x) - f_{\theta_t}(x)\|$ for any fixed $t$:** The proof for a fixed network on the path is almost the same as the proof for the end point. Instead of considering $\hat{x}^j_i$, now we consider $\hat{x}^j_i(t)$, which is the vector before activation at layer $j$ if the weights $A_2, \ldots, A_i$ are replaced by $A_2 + t(\hat{A}_2 - A_2), \ldots, A_i + t(\hat{A}_i - A_i)$. We can still use Lemma 5 to bound the noise produced by replacing the weight matrix at a single layer because

$$\|U(A_i + t(\hat{A}_i - A_i) - A_i)x\| = t\|U(\hat{A}_i - A_i)x\| \leq \|U(\hat{A}_i - A_i)x\|.$$

Thus, we can still use the above induction proof to show that for any fixed $0 \leq t \leq 1$, let $\epsilon' = O\left(\sqrt{\frac{pc^2 d^2 \log(mdh_d/\delta)}{h_{\min} \min_{2 \leq i \leq d} \mu_i^2 \mu_{i\rightarrow}^2}}\right)$, with probability at least $1/2 - \delta$,

$$\|f_\theta(x) - f_{\theta_t}(x)\| \leq \epsilon'\|f_\theta(x)\|.$$

**Bounding $\|f_\theta(x) - f_{\theta_t}(x)\|$ for every $t$:** Finally, we show that $\|f_\theta(x) - f_{\theta_t}(x)\|$ is bounded for every point on the path via an $\epsilon'$-net argument. Similar to previous steps, letting $\epsilon' = O\left(\sqrt{\frac{pc^2 d^2 \max_{x \in S}(\|f_\theta(x)\|^2) \log(mdh_d/\delta)}{h_{\min} \min_{2 \leq i \leq d}(\mu_i^2 \mu_{i\rightarrow}^2)}}\right)$, we know that with probability at least $1/2 - \delta$,

$$\|f_\theta(x) - f_{\theta_1}(x)\| \leq \epsilon'/2.$$

Next, we show that on the path, the network output is smooth in terms of the parameters. According to Algorithm 1, we know for any $2 \leq i \leq d$, we have $\|\hat{A}_i\| \leq 4\|A_i\|$, so $\|\hat{A}_i - A_i\| \leq 5\|A_i\|$. For any $2 \leq i \leq d$, let $A_{i,t} = A_i + t(\hat{A}_i - A_i)$. Note $\|A_{i,t}\| \leq (1-t)\|\hat{A}_i\| + t\|A_i\| \leq 4\|A_i\|$. For any $t, t'$ and any $2 \leq i \leq d$, let $\theta^i_{t,t'}$ be $\theta_t$ with the weight matrix at every layer $2 \leq j \leq i$ replaced by $(A^j + t'(\hat{A}^j - A^j))$. For convenience, we also denote $\theta_t$ as $\theta^1_{t,t'}$. Given $\tau < 1/2$, for any $\tau \leq t \leq 1 - \tau$ and for any $-\tau \leq \kappa \leq \tau$, we can bound $\|f_{\theta_{t+\kappa}}(x) - f_{\theta_t}(x)\|$ as follows:

$$\|f_{\theta_{t+\kappa}}(x) - f_{\theta_t}(x)\| \leq \sum_{2 \leq i \leq d} \|f_{\theta^i_{t,t+\kappa}}(x) - f_{\theta^{i-1}_{t,t+\kappa}}(x)\|$$

The output of layer $i - 1$ is the same for the two networks, of norm $\leq \|x\| \prod_{j=1}^{i-1}\|A_{j,t+\kappa}\|$. Hence the output of layer $i$ differs by at most $\kappa\|x\|\|\hat{A}_i - A_i\| \prod_{j=1}^{i-1}\|A_{j,t+\kappa}\|$ and the output differs by $\kappa\|x\|\|\hat{A}_i - A_i\| \prod_{j=1}^{i-1}\|A_{j,t+\kappa}\| \prod_{j=i+1}^{d}\|A_{j,t}\| \leq 5^d\kappa\|x\| \prod_{j=1}^{d}\|A_j\|$. Hence

$$\|f_{\theta_{t+\alpha}}(x) - f_{\theta_t}(x)\| \leq \sum_{2 \leq i \leq d} 5^d\|x\|\kappa \prod_{1 \leq j \leq d}\|A_j\|$$

$$\leq 5^d d\kappa\|x\| \prod_{1 \leq j \leq d}\|A_i\|.$$

Thus, given $\tau \leq \frac{\epsilon'}{2 \cdot 5^d d \max_{x \in S} \|x\| \prod_{1 \leq j \leq d} \|A_j\|}$, we know for any $\tau \leq t \leq 1 - \tau$ and for any $-\tau \leq \alpha \leq \tau$,

$$\|f_{\theta_{t+\alpha}}(x) - f_\theta(x)\| \leq \epsilon'/2. \tag{8}$$

There exists a set $Q = \{\theta_t\}$ with size $O(1/\tau)$ such that for any network on the path, the distance to the closest network in $Q$ is no more than $\tau$. If we can prove for any $\theta_t \in Q$, $\|f_\theta(x) - f_{\theta_t}(x)\| \leq \epsilon'/2$, we immediately know for any network $\theta_{t'}$ on the path $\|f_\theta(x) - f_{\theta_{t'}}(x)\| \leq \epsilon'$ by inequality (8).

By a union bound over $Q$, letting $\epsilon' = O\left( \sqrt{\frac{pc^2 d^2 \max_{x \in S}(\|f_\theta(x)\|^2) \log\left(\frac{mdh_d}{\delta\tau}\right)}{h_{\min} \min_{2 \leq i \leq d}(\mu_i^2 \mu_{i\to}^2)}} \right)$, we know with probability at least $1/2 - \delta$,

$$\|f_\theta(x) - f_{\theta_t}(x)\| \leq \epsilon'/2,$$

for any $\theta_t \in Q$.

Setting $\delta = 1/4$, we know there exists

$$\epsilon' = O\left( \sqrt{\frac{pc^2 d^3 \max_{x \in S}(\|f_\theta(x)\|^2) \log\left(\frac{mdh_d \max_{x \in S} \|x\| \prod_{1 \leq j \leq d} \|A_j\|}{\epsilon'}\right)}{h_{\min} \min_{2 \leq i \leq d}(\mu_i^2 \mu_{i\to}^2)}} \right)$$

such that with probability at least $1/4$,

$$\|f_\theta(x) - f_{\theta_t}(x)\| \leq \epsilon'$$

for any $x \in S$ and any $0 \leq t \leq 1$. Since the loss function is $\beta$-Lipschitz, we further know that for any $0 \leq t \leq 1$:

$$L(f_{\theta_t}) \leq L(f_\theta) + \beta\epsilon' = L(f_\theta) + \widetilde{O}(\sqrt{p}\epsilon).$$

$\square$

Now, we are ready to prove the main theorem.

**Theorem 2.** *Let $\theta^A$ and $\theta^B$ be two fully connected networks that are both $\epsilon$-noise stable, there exists a path with 10 line segments in parameter space $\pi : [0,1] \to \Theta$ between $\theta^A$ and $\theta^B$ such that[6] $L(f_{\pi(t)}) \leq \max\{L(f_{\theta^A}), L(f_{\theta^B})\} + \widetilde{O}(\epsilon)$ for $0 \leq t \leq 1$.*

**Proof of Theorem 2.** Setting dropout probability $p = 3/4$, by Lemma 5 and Lemma 3, if $h_{\min} = \widetilde{\Omega}(1)$, we know there exist $\theta_1^A$ and $\theta_1^B$ such that

1. in both networks, each weight matrix from layer 2 to layer $d$ has at least half of columns as zero vectors;

2. $L(f_{\theta_t^A}) \leq L(f_{\theta^A}) + \widetilde{O}(\epsilon)$ and $L(f_{\theta_t^B}) \leq L(f_{\theta^B}) + \widetilde{O}(\epsilon)$, for any $0 \leq t \leq 1$, where $\theta_t^A = \theta^A + t(\theta_1^A - \theta^A)$ and $\theta_t^B = \theta^B + t(\theta_1^B - \theta^B)$.

Since the dropout fraction in both $\theta_1^A$ and $\theta_1^B$ is at least half, we can connect $\theta_1^A$ and $\theta_1^B$ as we did in Lemma 2, while ensuring the loss doesn't exceed $\max\{L(f_{\theta^A}), L(f_{\theta^B})\} + \widetilde{O}(\epsilon)$. Connecting $\theta^A$ to $\theta_1^A$ and connecting $\theta_B$ to $\theta_1^B$ each take one line segment. By the construction in Lemma 2, connecting two dropped-out networks $\theta_1^A$ and $\theta_1^B$ takes 8 line segments. Thus, overall the path between $\theta^A$ and $\theta^B$ contains 10 line segments. $\square$

Next, we show that if there exists a "narrow" neural network achiving small loss, we can get a lower energy barrier using a smaller dropout probability.

**Theorem 3.** *Suppose there exists a network $\theta^*$ with layer width $h_i^*$ for each layer $i$ that achieves loss $L(f_{\theta^*})$, and minimum hidden layer width $h_{min}^* = \widetilde{\Omega}(1)$. Let $\theta^A$ and $\theta^B$ be two $\epsilon$-noise stable networks. For any dropout probability $1.5 \max_{1 \leq i \leq d-1}(h_i^*/h_i) \leq p \leq 3/4$, if for any $2 \leq i \leq d$, $1 \leq j \leq h_{i-1}$, $\|[A_i]_j\| = O(\sqrt{p})\|A_i\|_F$ then there exists a path with 13 line segments in parameter space $\pi : [0,1] \to \Theta$ between $\theta^A$ and $\theta^B$ such that $L(f_{\pi(t)}) \leq \max\{L(f_{\theta^A}) + \widetilde{O}(\sqrt{p}\epsilon), L(f_{\theta^B}) + \widetilde{O}(\sqrt{p}\epsilon), L(f_{\theta^*})\}$ for $0 \leq t \leq 1$.*

**Proof of Theorem 3.** Since $h_{\min} \cdot \max_{1 \leq i \leq d-1}(h_i^*/h_i) \geq h_{\min}^* = \widetilde{\Omega}(1)$, we have $h_{min} = \widetilde{\Omega}\left(\frac{1}{\max_{1 \leq i \leq d-1}(h_i^*/h_i)}\right)$. By Lemma 5 and Lemma 3, there exist $\theta_1^A$ and $\theta_1^B$ such that

1. in both networks, each weight matrix from layer 2 to layer $d$ has at least $h_i^*$ columns set to zero;

2. $L(f_{\theta_t^A}) \leq L(f_{\theta^A}) + \widetilde{O}(\sqrt{p}\epsilon)$ and $L(f_{\theta_t^B}) \leq L(f_{\theta^B}) + \widetilde{O}(\sqrt{p}\epsilon)$, for any $0 \leq t \leq 1$, where $\theta_t^A = \theta^A + t(\theta_1^A - \theta^A)$ and $\theta_t^B = \theta^B + t(\theta_1^B - \theta^B)$.

From the fact that at least $h_i^*$ units in layer $i$ of both $\theta_1^A$ and $\theta_1^B$ have been set to zero for $1 \leq i < d$—meaning that the corresponding rows of $A_i$ and columns of $A_{i+1}$ are zero—it follows from Lemma 2 that we can connect $\theta_1^A$ to an arbitrary permutation of $\theta^*$ using 8 segments while keeping the loss on the path no more than $\max\{L(f_{\theta_1^A}), L(f_{\theta^*})\}$. By choosing this permutation so that the non-zero units of $\theta^*$ do not intersect with those of $\theta_1^B$, we can then connect $\theta^*$ to $\theta_1^B$ using just 3 segments as done in the first step of our path construction in Lemma 2 seeing as there is no need to permute $\theta^*$ a second time. Combining these paths together with the paths that interpolate between the original parameters $\theta^A$ and $\theta^B$ and their dropout versions $\theta_1^A$ and $\theta_1^B$, we obtain a path in parameter space $\pi : [0,1] \rightarrow \Theta$ between $\theta^A$ and $\theta^B$ with 13 line segments such that $L(f_{\pi(t)}) \leq \max\{L(f_{\theta^A}) + \widetilde{O}(\sqrt{p}\epsilon), L(f_{\theta^B}) + \widetilde{O}(\sqrt{p}\epsilon), L(f_{\theta^*})\}$ for $0 \leq t \leq 1$. $\qquad\square$

## C  Proofs for disconnected modes in two-layer nets

**Proof of Theorem 4.** Define our loss over parameter space such that $L(f_\theta) = \frac{1}{n}\sum l(y_i, f_\theta(\mathbf{x}_i))$, where $\mathbf{x}_i \in \mathbb{R}^{h+2}$ is our $i^{\text{th}}$ data sample, $y_i \in \mathbb{R}$ the associated label, and $f_\theta(\mathbf{x}_i) = \mathbf{w}^T\phi(A\mathbf{x}_i)$ for $\theta = (\mathbf{w}, A) \in \mathbb{R}^{(h+2)\times h} \times \mathbb{R}^h$. We can represent the data samples as rows in a matrix $X \in \mathbb{R}^{n \times (h+2)}$—with $\mathbf{f}_i$ denoting the $i^{\text{th}}$ "feature" (i.e. column) of $X$—and the labels as elements of $\mathbf{y} \in \mathbb{R}^{n \times 1}$, as illustrated in Figure 4.

Choose $k, l, m, n$ such that $k < l < m < n$ where $k > h$, $l - k > h$, $m - l > 2$ and $n - m > h$.

When $i \leq l$, let

$$x_{i,j} = \begin{cases} i, & j = 1 \\ i - 1, & j = 2 \\ 1, & i \equiv j \pmod{h} \\ -1, & i \not\equiv j \pmod{h}, i \leq l \\ 0, & i \not\equiv j \pmod{h}, k < i \leq l. \end{cases}$$

When $l < i \leq m$, let

$$x_{i,j} = \begin{cases} -1, & j \leq 2, i \equiv j \pmod{2} \\ 0, & j \leq 2, i \not\equiv j \pmod{2} \\ 0, & j > 2, l < i \leq m. \end{cases}$$

When $i > m$, let

$$x_{i,j} = \begin{cases} 0, & j \leq 2 \\ -1, & j > 2, i \equiv j \pmod{h} \\ 0, & j > 2, i \not\equiv j \pmod{h}. \end{cases}$$

Finally, let $y_i = 1$ when $i \leq l$ and 0 otherwise.

From the fact that $\phi(\mathbf{f}_1) - \phi(\mathbf{f}_2) = \sum_{j=3}^{h+2}\phi(\mathbf{f}_j) = \mathbf{y}$ it follows that there exist networks with both two active hidden units and $h$ active hidden units that achieve minimal loss, with the former corresponding to the ground truth teacher network which generated our dataset.

Note in particular that the output layer weight corresponding to $\phi(\mathbf{f}_2)$ in the network with two active hidden units is negative, whereas in the network with $h$ active hidden units the output layer weights

$$
X = \begin{array}{c} \\ \mathbf{x}_1 \\ \mathbf{x}_2 \\ \vdots \\ \vdots \\ \mathbf{x}_k \\ \vdots \\ \mathbf{x}_l \\ \vdots \\ \vdots \\ \mathbf{x}_m \\ \vdots \\ \vdots \\ \mathbf{x}_n \end{array}
\begin{array}{cccccc}
\mathbf{f}_1 & \mathbf{f}_2 & \mathbf{f}_3 & \mathbf{f}_4 & \cdots & \mathbf{f}_{h+2} \\
1 & 0 & 1 & -1 & \cdots & -1 \\
2 & 1 & -1 & 1 & \ddots & \vdots \\
\vdots & \vdots & \vdots & \ddots & \ddots & -1 \\
\vdots & \vdots & -1 & \cdots & -1 & 1 \\
\vdots & \vdots & \vdots & \vdots & \vdots & \vdots \\
 & & & I_h & & \\
l & l-1 & \vdots & \ddots & \ddots & \ddots \\
 & -I_2 & & 0 & & \\
\vdots & \ddots & \vdots & \vdots & \vdots & \vdots \\
 & 0 & & -I_h & & \\
\vdots & \vdots & \vdots & \ddots & \ddots & \ddots
\end{array}
\qquad
\mathbf{y} = \begin{array}{c} y_1 \\ \vdots \\ \vdots \\ \vdots \\ \vdots \\ \vdots \\ y_l \\ \vdots \\ \vdots \\ \vdots \\ \vdots \\ y_n \end{array}
\begin{bmatrix} 1 \\ \vdots \\ \vdots \\ \vdots \\ \vdots \\ \vdots \\ 0 \\ \vdots \\ \vdots \\ \vdots \\ \vdots \\ \vdots \end{bmatrix}
$$

Figure 4: Our dataset.

are all positive. Thus, any path between the two networks must pass through a point in parameter space where at least one output layer weight is zero while the other $h - 1$ are positive. However, as shown in Lemma 6, there does not exist such a point in parameter space that achieves minimal loss. It follows that there exists a barrier in the loss landscape separating the original networks, both of which are global minima. Moreover, by adjusting $k$, $l$, and $m$ we can somewhat arbitrarily raise or lower this barrier. $\qquad\square$

**Lemma 6.** *There does not exist a set of $h - 1$ positive weights $w_i$ and vectors $\mathbf{h}_i \in \operatorname{span} X$ such that $\sum_{i=1}^{h-1} w_i \phi(\mathbf{h}_i) = \mathbf{y}$.*

*Proof.* We can think of each $\mathbf{h}_i$ as the output a particular hidden unit over all $n$ samples in our dataset and $w_i$ as the output layer weight associated to this hidden unit. We then have $\mathbf{h}_i = \sum a_{i,j} \mathbf{f}_j$, where the coefficients $a_{i,j}$ are elements of $A$.

First, if there did exist $w_i$ and $\mathbf{h}_i$ such that $\sum_{i=1}^{h-1} w_i \phi(\mathbf{h}_i) = \mathbf{y}$, then it must be the case for all $i$ that $\mathbf{h}_i = \sum a_{i,j} \mathbf{f}_j$ where $a_{i,j} \geq 0$ for all $j$. Otherwise, there would be non-zero elements in some $\mathbf{h}_i$ between indexes $l + 1$ and $n$ that would be impossible to eliminate in $\sum_{i=1}^{h-1} w_i \phi(\mathbf{h}_i)$ given that $w_i > 0$ for all $i$.

Second, any linear combination of $\mathbf{f}_1$ and $\mathbf{f}_2$ with positive coefficients would result in a vector whose first $l$ elements are positive and increasing. In contrast, the first $l$ elements of $Y$ are constant. And so from the fact that there does not exist $a_{i,j} > 0$ such that the first $l$ elements of $\sum a_{i,j} \mathbf{f}_j$ are decreasing—in particular because the first $k$ elements and next $l - k$ elements of $\sum_{j=3}^{h+2} a_{ij} x_j$ are periodic with length $h$—it follows that $a_{i,1}, a_{i,2} = 0$ for all $\mathbf{h}_i$.

Thus, we need only consider linear combinations of $\mathbf{f}_3$ through $\mathbf{f}_{h+2}$ with positive coefficients as candidates for $\mathbf{h}_i$. To this end, note that if a particular $\mathbf{f}_j$ has zero coefficient in all of $\mathbf{h}_1$ through $\mathbf{h}_{h-1}$, then $\sum_{i=1}^{h-1} w_i \phi(\mathbf{h}_i)$ will have zeros in every index congruent to $j \mod h$ and therefore cannot equal $\mathbf{y}$. Hence by the pigeonhole principle, in order to have $\sum_{i=1}^{h-1} w_i \phi(\mathbf{h}_i) = \mathbf{y}$ there must be some $i$ such that $\mathbf{h}_i = \sum_{j=3}^{h+2} a_{i,j} \mathbf{f}_j$ with at least two coefficients being non-zero. However, in any linear combination $\sum_{j=3}^{h+2} a_{i,j} \mathbf{f}_j$ where $a_{i,j}, a_{i,j'} > 0$ for at least two distinct $j, j'$, the elements in indexes

$k + 1$ to $l$ will be greater than the elements in indexes $1$ to $k$ that are congruent to $j \mod h$ and $j'$ $\mod h$. In contrast, the first $l$ elements of $\mathbf{y}$ are constant. Hence, similar to the case of $\mathbf{f}_1$ and $\mathbf{f}_2$, there cannot exist $\mathbf{h}_i = \sum_{j=3}^{h+2} a_{i,j} \mathbf{f}_j$ and positive coefficients $w_i$ such that $\sum_{i=1}^{h-1} w_i \phi(\mathbf{h}_i) = Y$. $\quad\square$

## D  Experimental details and further results

### D.1  Experimental details and hyperparameters

For all experiments on MNIST, we used a convolutional architecture consisting of 3 convolutional layers followed by a fully-connected output layer. Each convolutional layer consisted of $32$ $3 \times 3$ filters and used sufficient padding so as to keep the layer's output the same shape as its input. All networks were trained on an NVIDIA Tesla K20c GPU for $5000$ iterations with a batch size of $64$ using stochastic gradient descent with an initial learning rate of $0.1$ and a decay rate of $1\mathrm{E}^{-6}$. No significant hyperparameter tuning was applied. Images were normalized.

For the left and right plots in Figure 2, we report results averaged over 5 random trials and error bars corresponding to the standard deviation over these trials. For the center plot we simply computed the loss and accuracy over a linear path between a particular convolutional net and a single dropout version of itself. Specific to Figure 2, in applying dropout with probability $p$ we randomly sample a subset of $\lfloor 32(1-p) \rfloor$ units and rescale these units by $1/(1-p)$ while setting the remaining units to zero. In the left plot, each trial consisted of sampling 20 such dropout networks and reporting the performance of the network achieving the lowest loss. Losses and accuracies in all plots were computed on a random batch of $4096$ training images.

On CIFAR-10, we trained VGG-11 networks on an NVIDIA Titan X GPU for 300 epochs with SGD with a batch size of 128, with weight decay 5e-4, momentum 0.9, and an initial learning rate of 0.05 which is decayed by factor of 2 every 30 epochs. We used channel-wise dropout at all convolutional layers. The dropout rates are $p = 0.25$ at the first three layers and are $p = 0.5$ at the others. Ordinary dropout with $p = 0.5$ is used at every fully-connected layers except for the last one (the softmax layer).

### D.2  Straight interpolation between two models

As demonstrated in Figure 5, a straight line interpolation between two noise stable model may incur *large* losses and *poor* accuracies. The models are the same as used in Figure 3.

Figure 5: Loss and accuracy from directly interpolating between two noise stable models.

## D.3 Verification of noise stability conditions

### D.3.1 Layer cushion

### D.3.2 Interlayer cushion

### D.3.3 Activation contraction

### D.3.4 Interlayer smoothness

# E Tools

We use matrix concentration bounds to bound the noise produced by dropping out one single layer (Lemma 5).

**Lemma 7** (Matrix Bernstein; Theorem 1.6 in (Tropp, 2012)). *Consider a finite sequence $\{Z_k\}$ of independent, random matrices with dimension $d_1 \times d_2$. Assume that each random matrix satisfies*

$$\mathbb{E}[Z_k] = 0 \text{ and } \|Z_k\| \leq R \text{ almost surely.}$$

*Define*

$$\sigma^2 := \max \left\{ \Big\| \sum_k \mathbb{E}[Z_k Z_k^*] \Big\|, \Big\| \sum_k \mathbb{E}[Z_k^* Z_k] \Big\| \right\}.$$

*Then, for all $t \geq 0$,*

$$\Pr \left\{ \Big\| \sum_k Z_k \Big\| \geq t \right\} \leq (d_1 + d_2) \exp \left( \frac{-t^2/2}{\sigma^2 + Rt/3} \right).$$

As a corollary, we have:

**Lemma 8** (Bernstein Inequality: Vector Case). *Consider a finite sequence $\{v_k\}$ of independent, random vectors with dimension $d$. Assume that each random vector satisfies*

$$\|v_k - \mathbb{E}[v_k]\| \leq R \text{ almost surely.}$$

*Define*

$$\sigma^2 := \sum_k \mathbb{E} \big[ \|v_k - \mathbb{E}[v_k]\|^2 \big].$$

*Then, for all $t \geq 0$,*

$$\Pr \left\{ \Big\| \sum_k (v_k - \mathbb{E}[v_k]) \Big\| \geq t \right\} \leq (d+1) \cdot \exp \left( \frac{-t^2/2}{\sigma^2 + Rt/3} \right).$$