[Reviews · NeurIPS 2019]

Reviewer 1



The paper gives a clear overview of the “mode-connectivity” or epsilon-mode connectivity phenomenon in two- and arbitrary-layer neural networks, and how overparametrization has been studied as a potential explanation. Overparametrization is shown to enable dropout in a structural (rather than algorithmic) sense: if solutions are “dropout-stable”, then there exists a path between them (explicit construction provided) satisfying the eps-mode connectivity property (that the loss stays within eps of the larger of the two loss values). An equivalence between noise-stability and dropout-stability is then derived using a straightforward dropout algorithm, to extend and expand upon the results from dropout-stability to noise-stability. Experimental results show that the loss along paths as described for the minima studied is indeed nearly constant, and largely unaffected by dropout. Finally, a counterexample shows that overparametrization does not guarantee (eps-)mode connectivity. The paper is well motivated and generally well written, and the settings described are not restrictive. The path constructions are not overly complicated, but would benefit from being described more in math or pictorially (e.g., the block matrix multiplications where zeros propagate) rather than in paragraphs. Also, this is partly a function of space constraints, but there is much geometric meaning in the various “cushion” definitions that could be further touched upon (with greater concision elsewhere).

Reviewer 2



Although this paper tackles an interesting problem and has some theoretical results, I do not fully understand what the assumptions really mean and whether they are valid assumptions. I am also not very much convinced with the experimental results. I guess the main point of criticism for this paper is that, at least to me, the paper fails to justify the stability assumptions and to clearly articulate what these assumptions mean. First of all, the authors claim that dropout stability and noise stability is likely satisfied by the solutions obtained from gradient-based optimization, but there is not enough justification for that. The authors use these stability concepts from generalization literature, which is different from optimization. So, I am not convinced if these stability conditions are valid assumptions to make in the first place. I am also very confused with the assumptions themselves. - To start with dropout stability, it seems like only \theta_1 is used in the proof technique. Why does Definition 1 require all \theta_1, \theta_2, \dots, \theta_{d-1}? - The quantifier in Definition 1 looks incorrect. “There exists a subset of at most h_i/2 hidden units in each of the layers from i through d-1” means that each of the layer i, i+1, \dots, d-1, the subset size is at most h_i/2. Note that even for layer d-1, the subset size depends on h_i. Is this what you really meant? According to the proof of Lemma 2, I think the definition must change because it looks like subset for layer j must be at most cardinality h_j/2. - For the quantities in Definition 2, there isn’t enough explanation on what these quantities are capturing. For example, is it better (more stable) to have large mu or small mu? - Definition 3 doesn’t make sense to me, because the smoothness \rho is defined to be the smallest number but if you make it smaller and smaller, then the RHS of the inequalities will get larger and larger, which means that the infimum of such \rho must be 0. - Definition 4 also is difficult to interpret; the authors present some scary quantity without any good explanation. Especially, the authors say that the network is more robust if \epsilon is small. Then this means that the network is more robust if c is small, which means that the activation $\phi(x^i)$ must scale up all $x^i$. So this encourages $\phi$ to amplify $x^i$. However, I don’t understand why this leads to stability because if $\phi$ amplifies its input, then any noise will also be amplified by a big factor, leading to instability. - Connection of noise stability to drop stability is a bit weird as well; at least intuitively, it looks like a circular argument. Interlayer smoothness is defined using dropout noise (Alg 1) and then noise stability is defined using interlayer smoothness, and it is shown that noise stability implies dropout stability. I also have some comments about experimental results: - Why are experiments carried out on convolutional neural networks, when most of the theory developed is on fully connected neural networks? - The authors claim that Figure 2 shows that the solution is dropout stable. However, by Def 1 the dropout probability must be at least 0.5, and looks like the loss value 0.5 we get after dropout with p=0.5 doesn’t look like a very small number. So, I’m not fully convinced that the solutions that we get from training is dropout stable. - I think for the plots of loss function value through the constructed paths don’t give a good picture of whether the loss function value is indeed kept small along the path. Maybe comparing with direct linear interpolation will help illustrate the strength of the constructed path. --------------------------------------------- (post-rebuttal) I read the authors’ feedback and the other reviews. I think my negative impression of this paper came in part from some critical typos in the definitions (Def 1 and 3) which made me very confused while reading the paper. Thanks to the author response and the other reviewers, most of my concerns were well-addressed. However, I still believe the paper has some room for improvement, especially in its clarity of definition (and sketch of proofs, as the other reviewers pointed out). Besides correcting typos, I think it might be helpful to readers if the authors provided more explanation on the quantities defined in Definitions 2 and 4. At its current status, the paper relies heavily on readers’ prior knowledge of [Arora et al. 2018]; I believe the paper should be more self-contained. Re: purpose of \theta_2 through \theta_{d-1}: Now I understand that step (2) in Figure 1 uses \theta_2, thanks for clarifying. However, it was unclear from the main text that these points are used in the construction, because Lemma 1 and the following discussion only mentions \theta_1. I hope that this will be clarified in the next revision. Overall, I guess my initial score was overly harsh on this submission; I have updated my score.

Reviewer 3



Summary: This paper provides a theoretical analysis of the mode-connectivity phenomenon in deep learning [1,2]. First, under the assumption of stability to dropout (or more precisely existence of 1 dropout pattern under which the loss is stable, see Def. 1), the authors prove existence of a low-loss path consisting of a number of line segments that is linear in the number of layers. Under more restrictive noise stability assumptions [see 3] the authors prove the existence of paths with a constant number of linear segments. Next, the authors construct an example of architecture and dataset for which the mode connectivity doesn't hold despite overparameterization. Finally, the authors empirically evaluate the behavior of the noise stability metrics they introduced, and visualize train loss and accuracy along a path constructed by their method. Originality. To the best of my knowledge the present work provides first theoretical insights into mode connectivity for general deep neural networks. There exists related work on mode connectivity for restricted classes of neural nets [see e.g. 4], which is appropriately cited. Quality. The analysis is technically sound (I checked the details of proofs of Theorems 1,4, and skimmed the proofs for Theorems 2, 3). The constructions in the proofs are interesting and fairly intuitive. One of small issue I can see, is that in the definition if interlayer smoothness the authors work with logits of the networks obtained along the linear path from a given network to its dropout version. It seems like requiring interlayer smoothness could potentially be implicitly requiring the network to be stable along this linear path. However, the results are interesting nonetheless. Another question I have is about the construction of the dataset in appendix C. Do I understand correctly that for the case l < i ≤ m, it could happen that i ≠ j mod h for all l < i ≤ m, as you only require m - l > 2? In this case, the entries corresponding to this range of i's would all be zeros, and I believe this would break the proof of the fact that all a_{i,j} must be non-negative. Even if this is true, this is easy to fix by changing the condition i = j mod h to i = j mod 2. In the empirical part of the work, I find it hard to analyze the distributions of the noise stability notions, as they only appear in the asymptotic bounds. In particular, it's unclear to me what exactly should I understand from their specific values. Could the authors please comment on this? Clarity The paper is generally clearly written and easy to follow. There are several things that could be fixed to improve clarity however: 1. The difference between \Omega and O complexities is not explained. 2. Definition 1 was not very clear to me initially, I think it's vague the way it's written. It says that there exists a subset of [h_i / 2] neurpons in layers i to d, such that a certain condition holds. I believe, it should instead say something like for each i there exists solution theta^i such that for each i ≤ j ≤ d there are at most [h_j / 2] nonzero neurons on layer j. Lemma 2 is ambiguous in a similar way, it should probably say ...[hi / 2] of the units in each hidden layer i for each i... rather than ...[hi / 2] of the units in each hidden layer... 3. While the proof of Lemma 1 in the appendix is clear, it would help if the main text included a definition of Ls and Rs. Significance I believe this work is significant in the sense that it sheds new light on the mode connectivity phenomenon, and structure of minima in neural networks. In my opinion this paper would be a good contribution to NeurIPS, so I recommend an accept. [1] Loss Surfaces, Mode Connectivity, and Fast Ensembling of DNNs; Timur Garipov, Pavel Izmailov, Dmitry Podoprikhin, Dmitry Vetrov, Andrew Gordon Wilson [2] Essentially No Barriers in Neural Network Energy Landscape; Felix Draxler, Kambis Veschgini, Manfred Salmhofer, Fred A. Hamprecht [3] Stronger generalization bounds for deep nets via a compression approach S Arora, R Ge, B Neyshabur, Y Zhang [4] Topology and Geometry of Half-Rectified Network Optimization; C. Daniel Freeman, Joan Bruna *Post-Rebuttal* I have read the other reviews, and rebuttal. I was satisfied with the rebuttal and maintain my assessment.

[Author Response · NeurIPS 2019]

Thanks to all the reviewers for their thorough feedback and valuable suggestions.

**Reviewer 1**

We will add more intuitions and pictures to make the proofs in our appendix clearer.

**Reviewer 2**

*Typos*: We will fix the typos in Definitions 1 (i.e. $h_j/2$ for layer $j$) and 3 (i.e. largest number instead of smallest).

*Justifying our stability assumptions*: For neural networks trained with standard algorithms, noise stability was previously
observed in [Marcos et al. 2018, Arora et al. 2018]. While there is no formal proof showing any optimization algorithm
must find a noise stable solution, there is evidence that the solutions found are indeed noise stable. Our own experiments
also show that the paths constructed using these properties can indeed connect two different solutions.

*Purpose of $\theta_2$ through $\theta_{d-1}$*: Our path construction for Theorem 1 passes through each $\theta_i$, not just $\theta_1$. For example, in
Figure 1 Step (2) uses $\theta_2$ and Step (4) uses $\theta_1$. We describe how to connect each $\theta_i$ to $\theta_{i-1}$ in our proof of Lemma 1.
We will add some helpful figures in the revised version that should hopefully clarify to our path construction.

*Properties in Definition 2*: It is better to have larger $\mu$. This definition is exactly the same as in [Arora et al. 2018], due
to space limitations we chose to focus on properties that are slightly different.

*Definition 4, question about activation contraction $c$*: Main confusion here is that the noise stability quantities are
defined *specifically* for ReLU networks. For ReLU activations, the constant $c$ is mostly related to the fraction of neurons
that are active. The definitions would need to be different for other activations. While there are a lot of terms in
Definition 4, the overall message is that larger layer cushion and larger interlayer cushion leads to better noise stability
(i.e. smaller $\epsilon$). We also note that the theory is asymptotic, and that the numbers computed in a real network might be
loose; however our experimental results suggest that the theory does indeed lead to paths connecting different solutions.

*Explaining interlayer smoothness, "cirular" argument*: At a high level, interlayer smoothness assumes the network is
close to its *linear* approximation, but what we need to prove is that the network output is nearly *constant*. A large class
of functions have good linear approximations, but not all linear functions are constants. Only in combination with our
other noise stability assumptions can we show that the dropped out network has similar loss as the original network.

*Experiments on CNNs*: As explained in Remark 1, our dropout-based path constructions naturally extend to convolutional
nets since we can think of each channel as one hidden unit. All our noise stability properties also apply to convolutional
nets (as in [Arora et al. 2018]). We chose to show experiments with convolutional nets because these architectures are
widespread and of practical interest.

*Figure 2*: Figure 2 is meant to construct a path based on Theorem 3 (instead of Theorem 1 which requires dropping-out
1/2 the units). In Theorem 3 we show that if there exists a small network with low loss, then one only needs to drop out
a smaller fraction of the units in each hidden layer. The left plot shows the error under different levels of dropout; the
middle plot shows that connecting a network with its dropout version with $p = 0.2$ has almost constant loss; the right
plot shows that there exists a neural network whose size is $0.2$ times the original network that has low loss. Combining
these three plots with Theorem 3, we know that there exists a path with almost constant loss. We also note that data in
Figure 2 are from networks NOT trained with dropout. We will clarify these points in the final version.

*Comparison to interpolation*: Interpolating results in substantially higher loss than our path construction (loss/accuracy
1.61/67.2% on MNIST and 2.34/10% on CIFAR). We will add a plot in the revision. Thank you for the suggestion!

**Reviewer 3**

*Typos*: We will fix the typos in Definition 1 (same one pointed out by Reviewer 2), Lemma 2 and the construction of our
counterexample. Thanks for pointing these out!

*What we get from interlayer smoothness*: Interlayer smoothness does not in and of itself ensure that the output of the
network is stable as we discuss in our response to Reviewer 2. Requiring the property to hold for all $t$ is crucial for us
to be able to directly interpolate between the original network and the fully dropped out version using a single linear
segment. Note however that even if we only assume the property holds at the endpoint of this path we would still be
able to connect the original network to its dropped out version, but doing so would come at the cost of needing more
segments to construct the path as we did in Lemma 1.

*Figure 3*: The leftmost plot in Figure 3 is simply meant to give readers a sense for what the values for each of the
different components that comprise our ultimate definition of noise stability tend to look like in practice: the quantities
appearing in the denominator of noise stability are reasonable constants bounded away from zero, and conversely those
appearing in the numerator are not too large. It is true that the bounds are asymptotic and directly computing the bound
may not give a good number on real data, but our experiments show that the paths constructed are reasonable (although
they are not the same as the paths constructed in [Garipov et al. 2018, Draxler et al. 2018]).

[Meta-Review · NeurIPS 2019]

The paper provides a plausible first-cut theoretical explanation of mode connectivity in deep nets trained using gradient based optimization. The results are developed based on suitable assumptions on resilience to perturbations, in particular based on dropout training and noise stability. The paper also provides approaches to construct piece-wise linear paths to connect solutions under these assumptions. The reviewers liked the paper overall. There is agreement that the paper makes a good contribution, and also does not oversell the results. Some reviewers felt that the paper can be strengthened by improving clarity on certain definitions and ideas behind path constructions.